# High-resolution analyses of concentrations and sizes of refractory black carbon particles deposited in northwestern Greenland over the past 350 years – Part 2: Seasonal and temporal trends in refractory black carbon originated from fossil fuel combustion and biomass burning

**Kumiko Goto-Azuma**[1,2]**, Yoshimi Ogawa-Tsukagawa**[1]**, Kaori Fukuda**[1]**, Koji Fujita**[3]**, Motohiro Hirabayashi**[1]**, Remi Dallmayr**[1,a]**, Jun Ogata**[1]**, Nobuhiro Moteki**[4]**, Tatsuhiro Mori**[5]**, Sho Ohata**[6]**, Yutaka Kondo**[1]**, Makoto Koike**[7]**, Sumito Matoba**[8]**, Moe Kadota**[8]**, Akane Tsushima**[1,b]**, Naoko Nagatsuka**[1,c]**, and Teruo Aoki**[1]

[1]National Institute of Polar Research, Tachikawa, Tokyo, 190-8518, Japan
[2]SOKENDAI, Shonan Village, Hayama, Kanagawa, 240-0193, Japan
[3]Graduate School of Environmental Studies, Nagoya University, Nagoya, 464-8601, Japan
[4]Department of Chemistry, Graduate School of Science, Tokyo Metropolitan University, Hachioji, Tokyo, 192-0397, Japan
[5]Department of Applied Chemistry, Faculty of Science and Technology, Keio University, Yokohama, Kanagawa, 223-8522, Japan
[6]Institute for Space-Earth Environmental Research, Nagoya University, Nagoya, 464-8601, Japan
[7]Department of Earth and Planetary Science, Graduate School of Science, University of Tokyo, Bunkyo-ku, Tokyo, 113-0033, Japan
[8]Institute of Low Temperature Science, Hokkaido University, Sapporo, 060-0819, Japan
[a]now at: Alfred Wegener Institute for Polar and Marine Research, Bremerhaven, Germany
[b]now at: Graduate School of Integrated Science and Technology, Nagasaki University, Nagasaki, 852-8521, Japan
[c]now at: Japan Agency for Marine-Earth Science and Technology, Yokosuka, Kanagawa, 237-061, Japan

**Correspondence:** Kumiko Goto-Azuma (kumiko@nipr.ac.jp)

**Abstract.** The roles and impacts of refractory black carbon (rBC), an important aerosol species affecting Earth's radiation budget, are not well understood owing to a lack of accurate long-term observations. To study the temporal changes in rBC since the pre-industrial period, we analyzed rBC in an ice core drilled in northwestern Greenland. Using an improved technique for rBC measurement and a continuous flow analysis (CFA) system, we obtained accurate and high-temporal-resolution records of rBC particle size and mass/number concentrations for the past 350 years. Number and mass concentrations, which both started to increase in the 1870s associated with the inflow of anthropogenically derived rBC, reached their maxima in the 1910s–1920s and then subsequently decreased. Backward-trajectory analyses suggest that North America was likely the dominant source region of the anthropogenic rBC in the ice core. The increase in anthropogenic rBC shifted the annual concentration peaks of rBC from summer to winter–early spring. After rBC concentrations diminished to pre-industrial levels, the annual peak concentration of rBC returned to the summer. We found that anthropogenic rBC particles were larger

than biomass burning rBC particles. By separating the rBC in winter and summer, we reconstructed the temporal variations in rBC that originated from biomass burning, including the period with large anthropogenic input. The rBC that originated from biomass burning showed no trend in increase until the early 2000s. Finally, possible albedo reductions due to rBC are discussed. Our new data provide key information for validating aerosol and climate models, thereby supporting improved projections of future climate and environment.

## 1   Introduction

Black carbon (BC) particles, which are emitted by incomplete combustion of biomass and fossil fuels, affect Earth's radiation budget and thus climate (Bond et al., 2013; Moteki, 2023; Matsui et al., 2022; Legrand et al., 2016). In turn, changes in climate can affect emissions of BC from biomass burning through natural processes such as wildfires. Global warming is considered a major cause of the recent increase in large wildfires globally that can cause serious damage to ecosystems and human society (Calkin et al., 2023; Keeley and Syphard, 2021; Wang et al., 2021; Keane et al., 2008). Increased occurrence of large wildfires in the future could affect Earth's radiation budget and change the frequency at which certain regions are exposed to serious hazard. Increases in fossil fuel combustion since the Industrial Revolution have changed Earth's radiation budget and contributed to the warming or cooling over the past century (Shindell and Faluvegi, 2009; McConnell et al., 2007; Breider et al., 2017). To understand the effects of BC on the radiation budget and the impact of climate change on BC emissions, the long-term changes in the concentrations and size distributions of BC particles should be known. Data obtained since the pre-industrial period are particularly valuable because we cannot fully understand the anthropogenic effects without characterizing BC in a pristine environment. The Arctic is the key region for clearer elucidation of the roles of BC because the Arctic has warmed at a rate 4 times faster than that of the global average over the past half-century, leading to drastic changes such as sea ice retreat, enhanced glacier mass loss, and ecosystem changes (Rantanen et al., 2022). Despite numerous studies based on observations and aerosol/climate models (e.g., Bond et al., 2013, and references therein), we have only limited knowledge on BC owing to the lack of accurate long-term in situ observations (Mori et al., 2019). For the Arctic region, data are particularly sparse and few long-term records of BC size distribution exist.

Although there have been no direct observations before the past few decades, ice cores drilled in the Arctic have provided long-term records of BC. Development of the Single-Particle Soot Photometer (hereafter, SP2; Droplet Measurement Technologies, USA) (Stephens et al., 2003; Baumgardner et al., 2004) enabled measurements of refractory BC (rBC), the terminology used for incandescence-based BC measurements (Petzold et al., 2013; Lim et al., 2014), in Arctic ice cores, where BC concentrations are low and sample volumes are limited (McConnell et al., 2007; Zdanowicz et al., 2018; Zennaro et al., 2014; Osmont et al., 2018). A continuous flow analysis (CFA) system is often used with the SP2 for high-resolution analysis of ice cores (McConnell et al., 2007; Zdanowicz et al., 2018; Zennaro et al., 2014). With an SP2 attached to a CFA system, McConnell et al. (2007) reconstructed rBC mass concentrations since the pre-industrial period in central and southern Greenland. They showed that rBC concentration began to gradually rise after 1850, followed by a rapid increase around 1890, a peak at around 1910, and then an erratic decline through the late 1940s and a sharp drop in the 1950s. They attributed the increase to rBC derived mainly from fossil fuel combustion in North America. Similar anthropogenic temporal trends have been reported for other Greenland sites (McConnell, 2010). The rBC flux records presented by McConnell (2010) suggest that the anthropogenically derived increase in rBC was substantially less in northern Greenland than in southern Greenland, which is closer to the emission sources in North America and western Europe. McConnell et al. (2007) also reported that the greatest increase in anthropogenic rBC occurs in winter. However, no rBC particle size data from Greenland ice cores have been published to date.

At Arctic sites outside Greenland, only a few ice cores have been analyzed for BC. An ice core from Holtedahlfonna (Svalbard) indicated that BC mass concentration started to increase after 1850 and peaked around 1910, similar to the rBC record of ice cores from Greenland (Ruppel et al., 2014). BC concentrations in the Holtedahlfonna core increased again between 1970 and 2004, reaching unprecedented values in the 1990s. This increase is not seen in Greenland ice cores, and it contradicts atmospheric BC observations from Svalbard and other Arctic sites (Sharma et al., 2013), which indicate declining concentrations of atmospheric BC. Ruppel et al. (2014) attributed the differences partly to the different sources of anthropogenic BC affecting Svalbard and Greenland attributable to different air mass trajectories. They also suggested that changes in scavenging efficiency might have affected the Holtedahlfonna BC record. An ice core from Lomonosovfonna, another site in Svalbard (Osmont et al., 2018), showed a gradual increase in rBC during 1800–1859, followed by a dramatic increase from 1860. The concentrations displayed two maxima at around 1870 and 1895 before they started to decline. Between 1910 and 1949, concentrations of rBC were low. In contrast to the concentrations of rBC in Greenland, another notable increase was evident in

the Lomonosovfonna core after 1940, and the concentrations were at their highest in the 1950s and 1960s. The rBC concentrations started to decrease in the 1970s, i.e., much later than the start of the decline in Greenland. The authors argue that the differences between Greenland and Lomonosovfonna are partly related to the different source regions of the air masses reaching Greenland and Svalbard.

The differences between the Holtedahlfonna and Lomonosovfonna records might also reflect different methods used for the measurement of BC mass concentration. The samples from the Holtedahlfonna ice core were filtered, and then the filters were analyzed for BC using a thermal–optical method (Osmont et al., 2018), whereas the Lomonosovfonna and Greenland ice cores were analyzed using an SP2. Uncertainties regarding the filtering efficiency (Ruppel et al., 2014) and the effects of dust particles on the thermal–optical method could partly explain the differences in the long-term trends in BC concentrations. Furthermore, melt–freeze cycles that commonly occur at ice-coring sites in Svalbard would have affected the rBC concentrations (Osmont et al., 2018). Moreover, melt–freeze cycles could have agglomerated the rBC particles to larger sizes beyond the detection range of an off-the-shelf SP2 (Osmont et al., 2018; Wendl et al., 2014). An ice core rBC record from the Devon Ice Cap in the Canadian Arctic was also found to differ from the records of Greenland ice cores (Zdanowicz et al., 2018). Although such differences could be partly attributable to different rBC source regions, melt–freeze cycles could also have affected the Devon Ice Cap record. To investigate whether melt–freeze cycles did affect the derived rBC concentrations, we need to know the sizes of the rBC particles.

Even for ice cores drilled at sites where summer melting seldom occurs, such as those from interior Greenland, it is important to investigate the size distributions of rBC particles to verify whether they are within the detection range of the SP2 instrument. This is because the sizes of rBC particles in snow are often larger than those in the atmosphere (Schwarz et al., 2013; Mori et al., 2019) and exceed the detection range of a traditional SP2. The upper limit of detectable rBC diameter is usually approximately 500 nm for the off-the-shelf SP2; that for SP2 modified by Moteki and Kondo (2010) is approximately 850 nm (Moteki and Kondo, 2010; Mori et al., 2019), and that for the off-the-shelf SP2 Extended Range (SP2-XR) is 800 nm. If a large proportion of rBC particles have a diameter of $> 500$ or 850 nm, the BC mass concentrations would be underestimated (Mori et al., 2019; Goto-Azuma et al., 2024a). Furthermore, if an ultrasonic nebulizer, such as the U5000AT (CETAC Technologies, USA), was used with an off-the-shelf SP2, as was the case in most previous studies of rBC in ice cores (McConnell et al., 2007; Kaspari et al., 2011; Zennaro et al., 2014; Bisiaux et al., 2012a, b; Wang et al., 2015; Zdanowicz et al., 2018; Du et al., 2020), there would be large uncertainties in rBC mass concentrations (Wendl et al., 2014; Goto-Azuma

et al, 2024a). Because the nebulizing efficiency of this type of nebulizer varies markedly for rBC particles $< 850$ nm (Ohata et al., 2013; Mori et al., 2016; Goto-Azuma et al., 2024a), variation in efficiency should be considered when calculating accurate mass concentrations and size distributions (Ohata et al., 2013). However, this was not taken into account in most previous ice core studies. It is therefore important to analyze Arctic ice cores using an instrumental setup that allows the detection of rBC particles with a diameter of $> 850$ nm and also to consider the size-dependent efficiency of the nebulizer. We developed a CFA system that includes an rBC unit, which allows accurate high-resolution measurement of concentrations and size distributions of rBC particles with a diameter between 70 nm and 4 µm. Using this system, we analyzed an ice core drilled at the SIGMA-D site in northwestern Greenland. The details of this new system and its performance are described in the companion paper (Goto-Azuma et al., 2024a). In this study, we analyzed the data and investigated the temporal variability in the concentration and size distribution of rBC that originated from fossil fuel combustion and biomass burning.

The BC detected in Arctic ice cores, together with $NH_4^+$ and specific organic materials (i.e., formate, levoglucosan, vanillic acid, and $p$-hydroxybenzoic acid), has been used to reconstruct past biomass burning (Ruppel et al., 2014; Zennaro et al., 2014; Grieman et al., 2017, 2018; Fischer et al., 2015; Pokhrel et al., 2020; Legrand et al., 2016; Parvin et al., 2019). Although both BC and $NH_4^+$ have sources other than biomass burning (Osmont et al., 2018), levoglucosan, vanillic acid, and $p$-hydroxybenzoic acid primarily originate from biomass burning. However, the data regarding such organic materials usually have a lower temporal resolution than rBC and $NH_4^+$ data owing to the limitations of the measurement techniques. Furthermore, little is known about their changes during atmospheric transport and postdepositional processes (Hennigan et al., 2010). Different ice core proxies often show different temporal and spatial trends in biomass burning activities (Legrand and De Angelis, 1996; Legrand and Mayewski, 1997; Legrand et al., 1992, 2016; Kawamura et al., 2012; Grieman et al., 2017, 2018; Rubino et al., 2015; Zennaro et al., 2014). Compared with the Global Charcoal Database, which has widely been used to investigate changes in biomass burning on centennial to orbital timescales (Power et al., 2010; Marlon et al., 2016), ice core proxy records usually have higher temporal resolution. Even monthly or seasonally resolved continuous records of rBC and $NH_4^+$ for the past few centuries, millennia, and 100 000 years have been derived from several Arctic ice cores, thereby allowing the detection of high spikes in concentration in summer attributable to large boreal forest fires in northern North America and/or Siberia (Fischer et al., 2015; Zennaro et al., 2014). However, previous studies using rBC as a biomass burning tracer have been restricted to the pre-industrial period. This is because rBC originated from fossil fuel combustion contributed greatly to the total

rBC concentrations and obscured the temporal trends in rBC related to biomass burning.

In this study, we reconstructed approximately monthly resolved concentrations and sizes of rBC particles in northwestern Greenland over the past 350 years. The amounts of rBC originating from biomass burning and from fossil fuel combustion were distinguished owing to their different seasonal variability. In this paper, we discuss the temporal trends in the concentration and size of rBC particles originating from both sources, and we investigate the rBC source regions based on backward-trajectory analyses. We then estimate the potential albedo reductions based on the monthly mean rBC concentration data.

## 2 Materials and methods

### 2.1 Ice core processing, analyses, and dating

A 222.7 m ice core was drilled at the SIGMA-D site (77.636° N, 59.120° W; 2100 m a.s.l.; Fig. 1) in northwestern Greenland in spring 2014 (Matoba et al., 2015). The annual mean air temperature and accumulation rate at the site were estimated to be $-25.6\,°C$ and $0.23\,\text{w.e.}\,\text{yr}^{-1}$, respectively (Nagatsuka et al., 2021). We examined the melt features (ice layers and thin crusts) in the uppermost 20 m of the SIGMA-D ice core, where increased summer melting would be expected due to recent warming. We observed ice layers, with a maximum thickness of 10 mm, at only three depths. The 20 m average melt feature percentage (MFP) was 0.47 %. The maximum MFP per meter was 1.7 %, and 10 out of the 20 m had no melt features. Thus, the effects of melt–refreeze cycles are minimal at the SIGMA-D site.

The details of the ice core processing and analyses are described in the companion paper (Goto-Azuma et al., 2024a); therefore, we provide only a brief summary here. The top 175.77 m of the core was divided into two vertical sections (Sections A and B) in the field. Section A was kept frozen and transported to the National Institute of Polar Research (NIPR) in Japan; Section B was cut, melted, and bottled in the field.

Down to the depth of 112.87 m in Section A, we cut quadrangular prism samples with a cross-section of $34\,\text{mm} \times 34\,\text{mm}$. For the depth interval between 6.17 and 112.87 m, we analyzed rBC, stable isotopes of water, and six elements (i.e., $^{23}$Na, $^{24}$Mg, $^{27}$Al, $^{39}$K, $^{43}$Ca, and $^{56}$Fe) using the NIPR CFA system. The top 6.17 m of Section A was too fragile to be analyzed using the CFA system; hence, we manually cut it into segments of approximately 0.1 m. These "discrete samples" were decontaminated in a $-20\,°C$ cold room using a precleaned ceramic knife and then placed in powder-free plastic bags. They were then melted and transferred to precleaned glass and polypropylene bottles in a class 10 000 clean room. The samples in glass bottles were analyzed for stable isotopes of water and rBC. For the discrete samples, stable isotopes of water were analyzed using a near-infrared

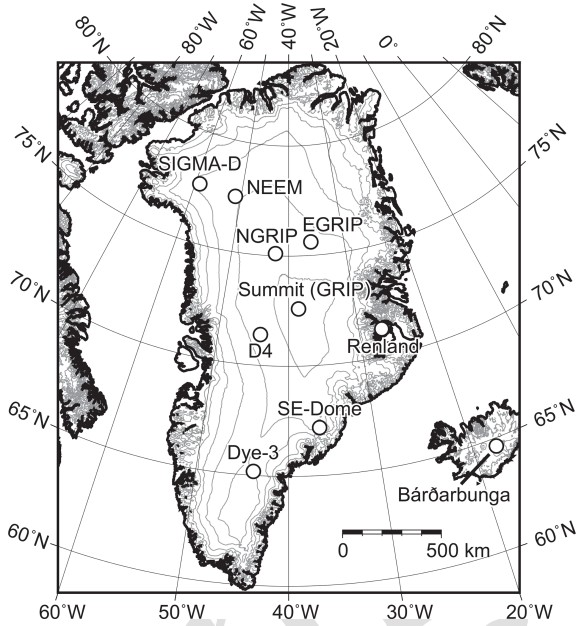

**Figure 1.** Location of the SIGMA-D site and other drill sites.

cavity ring-down spectrometer (L2120-i, Picarro Inc., USA), a high-precision vaporizer (A0211, Picarro Inc., USA), and an autosampler (PAL HTC9 – xt – LEAP, LEAP Technologies, USA). The precision of determination was $\pm 0.05\,‰$ for $\delta^{18}$O. The samples in the polypropylene bottles were analyzed for six elements (i.e., $^{23}$Na, $^{24}$Mg, $^{27}$Al, $^{39}$K, $^{40}$Ca, and $^{56}$Fe) using an inductively coupled plasma mass spectrometer (7700 ICP-MS, Agilent Technologies, USA) in a class 10 000 clean room at NIPR.

Both the CFA samples and the discrete samples were analyzed for rBC using the wide-range SP2 (Mori et al., 2016), which is a modified version of the SP2 (Droplet Measurement Technologies, USA), and a concentric pneumatic nebulizer (Marin-5, Teldyne CETAC, USA). The combination of the wide-range SP2 and the pneumatic nebulizer enabled us to extend the range of the size of rBC particles analyzed (70 nm < diameter < 4 μm) to beyond those of the off-the-shelf SP2 (70 nm < diameter < 500 nm) and the modified SP2 (Moteki and Kondo, 2010) or the off-the-shelf SP2-XR (50–70 nm < diameter < 800–850 nm). This combination and a careful calibration procedure enabled us to measure not only the concentration but also the diameter of rBC particles. The analytical errors of the rBC mass and number concentrations were estimated to be < 16 % (Mori et al., 2016; Goto-Azuma et al., 2024a). The reproducibility of the rBC number and mass concentrations for repeated measurements was usually better than 10 % (Mori et al., 2019; Goto-Azuma et al., 2024a). The detection limits of the rBC number and mass concentrations were approximately $0.35\,\text{counts}\,\mu\text{L}^{-1}$ and $0.02\,\mu\text{g}\,\text{L}^{-1}$, respectively.

Depths of Section B above 61.2 m were analyzed for $Na^+$, $K^+$, $Mg^{2+}$, $Ca^{2+}$, $Cl^-$, $NO_3^-$, and $SO_4^{2-}$ using two ion chromatographs (ICS-2100, Thermo Fisher Scientific, USA) at Hokkaido University (Japan), whereas depths between 61.2 and 112.87 m were analyzed for $NH_4^+$, $Na^+$, $K^+$, $Mg^{2+}$, $Ca^{2+}$, $Cl^-$, $NO_3^-$, and $SO_4^{2-}$ using two ion chromatographs (ICS-2000, Thermo Fisher Scientific, USA) at NIPR. The limit of detection of $Na^+$ measured at Hokkaido University was $10\,\mu g\,L^{-1}$, whereas that measured at NIPR was $0.2\,\mu g\,L^{-1}$. Stable isotopes of water were analyzed for all samples from Section B using a near-infrared cavity ring-down spectrometer (L2130-i, Picarro, USA) and a high-throughput vaporizer (A0212, Picarro, USA) at Hokkaido University. The precision of determination was $\pm 0.1\,‰$ for $\delta^{18}O$. For dating purposes, tritium concentrations were measured using a liquid scintillation counter (LSC-LB3, Aloka Co. Ltd., Japan) at 0.05 m intervals for the depth interval 19.15–26.47 m (Nagatsuka et al., 2021).

Figure 2 shows the seasonal variability in Na and $Na^+$ concentrations, together with that in $\delta^{18}O$. Concentrations of Na and $Na^+$ show maxima in winter and minima in summer, whereas the $\delta^{18}O$ shows maxima in summer and minima in winter (Nagatsuka et al., 2021; Legrand and Mayewski, 1997; Mosher et al., 1993). We dated Section B of the SIGMA-D core by annual layer counting using mainly $Na^+$ (Nagatsuka et al., 2021), which exhibited clearer seasonal variation compared to $\delta^{18}O$ and other ionic species. The seasonal variation in $\delta^{18}O$, typically used for annual layer counting, was often obscured by diffusion in the SIGMA-D core. However, we supplementarily used $Ca^{2+}$ and $\delta^{18}O$ data when annual peaks of $Na^+$ were not clearly observed. Additionally, we used a tritium peak (1963) and volcanic $SO_4^{2-}$ peaks (Katmai, 1912; Tambora, 1816; unknown, 1810; and Laki, 1783) as reference horizons, as reported by Nagatsuka et al. (2021). Because the CFA data from Section A and the discrete data from Section B agreed well (Fig. 2), we basically adopted the chronology of Section B for that of Section A with a few minor adjustments. The uncertainties of dating were estimated to be less than $\pm 2$ years. The CFA data covered 1653–2002, and the data from the top 6.17 m covered the period 2003–2013.

We divided 1 year into 12 months based on the assumption that the annual maxima and minima of $Na^+$ concentration correspond to 1 January and 1 July, respectively (Fig. 2). Each depth interval corresponding to a half year was evenly divided into 6 months. Using the CFA data, we calculated the annual mean and the monthly mean values of the number and mass concentrations of rBC particles. It is important to note that the months defined in this study may not align exactly with calendar months for the following reasons: (1) precipitation is not evenly distributed throughout the year (Figs. A1 and A2); (2) the minima and maxima of $Na^+$ concentrations do not necessarily coincide with 1 January and 1 July, respectively; and (3) dry deposition would have a

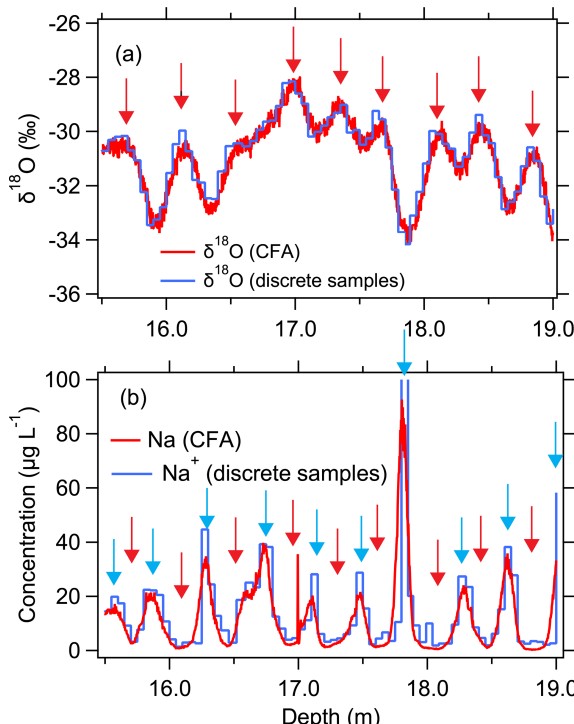

**Figure 2.** (a) $\delta^{18}O$ and (b) Na (and $Na^+$) concentrations. Red and blue curves represent data obtained from CFA measurements (this study) and discrete sample measurements (Nagatsuka et al., 2021) of the SIGMA-D core, respectively. Blue and red arrows indicate winter and summer, respectively. Winter and summer peaks were assumed to represent 1 January and 1 July of each year, respectively.

small contribution. As a result, there is inherent uncertainty in this definition. The discrepancy between the "months" defined in this study and actual calendar months could be 1 or 2 months. Nevertheless, we will refer to these periods as "months" hereafter.

As demonstrated in the companion paper (Goto-Azuma et al., 2024a), the dispersion lengths of the CFA system are $\sim 35$ and $\sim 39$ mm for Na and BC, respectively. However, we could usually resolve two peaks which were 10 mm apart, although the signal dispersion might slightly reduce the heights of the seasonal peaks.

## 2.2 Backward-trajectory analysis

To estimate the contributions of different air masses affecting the SIGMA-D and D4 sites (Fig. 1), we performed 10 d backward-trajectory analyses for the period 1958–2015. Dividing the globe into 21 regions (Fig. A3), we calculated the contribution from each region. We used the Hybrid Single-Particle Lagrangian Integrated Trajectory (HYSPLIT) model developed by the National Oceanographic and Atmospheric Administration (NOAA) (Stein et al., 2015). The initial air mass was set at three elevations at each site (i.e., 500, 1000, and 1500 m above ground level), and the accumulated prob-

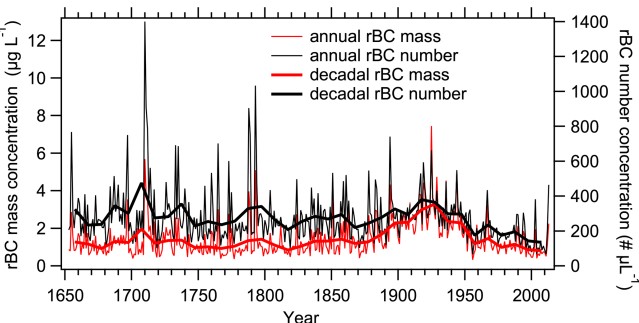

**Figure 3.** Annual mean (thin curves) and decadal mean (thick curves) concentrations of rBC. Red and black curves represent mass and number concentrations, respectively.

ability of the air mass in each 1° grid cell was calculated. Assuming wet deposition of rBC at the ice core sites, the air mass probability was weighted with the local daily precipitation; i.e., if no precipitation occurred, the air mass was not considered and vice versa. We used ERA5 precipitation data produced by the European Centre for Medium-Range Weather Forecasts (Hersbach et al., 2020). Details of the procedures are described in previous studies (Parvin et al., 2019; Nagatsuka et al., 2021).

For comparison, we also calculated the regional contributions not weighted for precipitation while accounting for dry deposition. However, we anticipate that dry deposition contributes only minimally compared to wet deposition (Appendix B).

## 3   Results and discussion

### 3.1   Impacts of anthropogenic emissions on long-term trends in concentrations and sizes of rBC particles

Figure 3 displays annual and decadal averages of number and mass concentrations of rBC during the past 350 years. Notably, monthly mean values could have been affected by the values of the adjacent 1–2 months, considering the resolution of the CFA data (Sect. 2.1 and Goto-Azuma et al., 2024a). Because we melted the core from the bottom to the top, the data for 1–2 months after large rBC concentration peaks could have been affected. However, the annual and decadal averages were unaffected by the CFA signal dispersion. We also calculated the annual rBC mass flux using annual mean rBC mass concentration data and annual accumulation rate data (Fig. A4). Since there are no long-term trends in annual accumulation rates, the temporal trends in rBC mass concentrations and rBC mass fluxes are consistent. Therefore, we use concentration data in the following discussion.

Many of the very high mass and number concentration peaks, such as those in 1710, originated from large boreal forest fires, as discussed in Sect. 3.3. Apart from these sporadic sharp peaks in number and mass concentrations,

their background levels started to increase in the 1870s, reached their maxima in the 1910s–1920s, and decreased again since the 1930s. Breakpoint analysis (Muggeo, 2003; de la Casa and Nasello, 2010) confirmed the timing of these increases and decreases. In the 1960s, rBC number and mass concentrations returned to their pre-industrial levels. In the 1980s and 1990s, number concentrations were below the pre-industrial level, whereas mass concentrations were similar to those of the pre-industrial level. Before 1850, the major sources of rBC in Greenland were likely to have been biomass burning emissions from boreal forest fires (Legrand et al., 2016; McConnell et al., 2007; Zennaro et al., 2014). The increases in rBC concentrations that occurred in the late 19th century to mid-20th century are likely attributable to inflow to Greenland of rBC of anthropogenic origin, as reported previously (McConnell, 2010; McConnell et al., 2007).

Direct comparison between the rBC concentrations in the SIGMA-D core and those in other Greenland ice cores is not strictly feasible owing to methodological differences. rBC measurements in other Greenland ice cores were conducted using the off-the-shelf SP2 coupled with an ultrasonic nebulizer (McConnell et al., 2007; McConnell, 2010; Zennaro et al., 2014). This setup allows the measurement of rBC particles with a diameter of less than 500 nm (Goto-Azuma et al., 2024a). In contrast, the measurements of the SIGMA-D ice core could detect rBC particles with a diameter up to 4 μm. Therefore, rBC concentrations in other Greenland ice cores might have been underestimated during periods when the diameter of large proportions of rBC particles exceeded approximately 500 nm. Owing to a lack of information on size distributions, the extent of the underestimation for other Greenland ice cores remains unknown. As described in the companion paper (Goto-Azuma et al., 2024a), if the off-the-shelf SP2 used in the previous studies had also been used for the SIGMA-D ice core, the extent of underestimation would have depended on depth and hence on age. However, the general temporal trends in annual mean rBC concentrations at the SIGMA-D site did not change notably if rBC particles with a diameter of > 500 nm, the maximum measurable diameter of the off-the-shelf SP2, were excluded (Fig. 4). Therefore, it is informative to compare the rBC concentration trends at the SIGMA-D site with those of other Greenland sites.

The long-term trends in rBC mass concentrations at the SIGMA-D site are broadly similar to those at other ice core sites in Greenland (McConnell et al., 2007; McConnell, 2010), including the D4 site. However, the SIGMA-D core shows much lower anthropogenic rBC concentrations, a later peak period, and a later onset of the reductions in comparison with those of the D4 core (Fig. 4). This is in accordance with the studies by McConnell et al. (2007) and McConnell (2010), which indicate that more southerly sites generally show higher anthropogenic rBC concentrations, an earlier peak period, and an earlier onset of the decline in an-

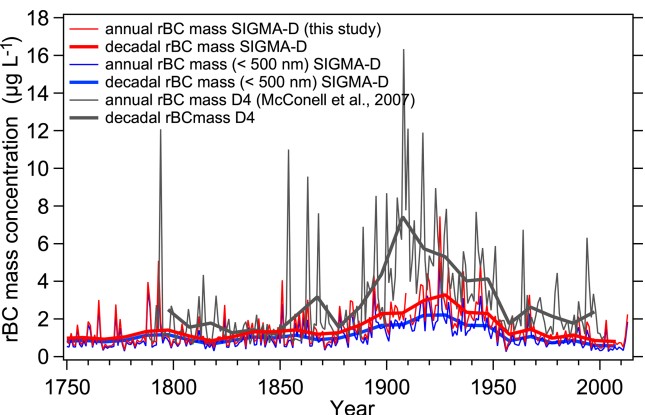

**Figure 4.** Annual mean (thin curves) and decadal mean (thick curves) mass concentrations of rBC. Red and black curves represent rBC concentrations at the SIGMA-D and D4 sites, respectively. Blue curves show rBC concentrations for rBC particles with a diameter of < 500 nm.

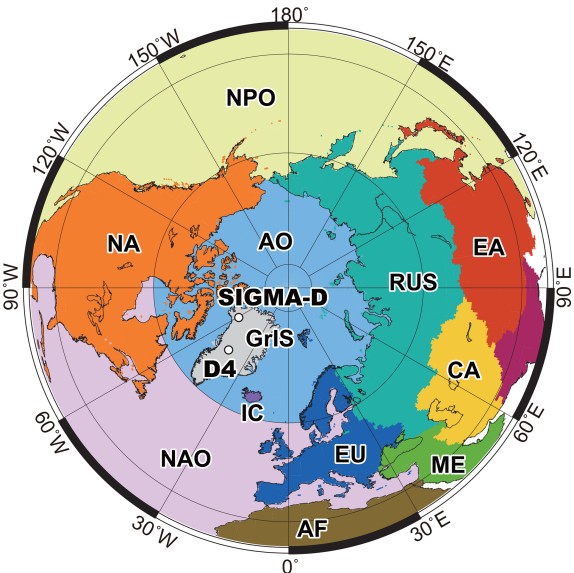

**Figure 5.** The 12 regions showing non-zero contributions in precipitation-weighted backward-trajectory analyses (GrIS: Greenland Ice Sheet. NA: North America. EU: Europe. RUS: Russia. CA: central Asia. EA: eastern Asia. ME: Middle East. AF: Africa. NPO: North Pacific Ocean. NAO: North Atlantic Ocean. AO: Arctic Ocean. IC: Iceland.).

thropogenic rBC concentrations in comparison with those of more northerly sites. The BC emission inventories for potential BC source regions indicate that emissions of anthropogenically derived BC peaked earlier in North America than in Europe and that the decline in anthropogenic BC concentrations started earlier in North America than in Europe or the former USSR (Osmont et al., 2018). The emission inventories used by Osmont et al. (2018) were those adopted for the Coupled Model Intercomparison Project phase 5 (CMIP5; Bauer et al., 2013; Eckhardt et al., 2023; Lamarque et al., 2010). If those emission inventories are reliable, then the slight difference in the temporal trends of rBC concentrations would indicate that southern Greenland sites (e.g., the D4 site) had been influenced mainly by anthropogenic emissions from North America, whereas northern Greenland sites (e.g., the SIGMA-D site) had been influenced by anthropogenic emissions from Europe and the former USSR (in addition to those from North America), as was the case for anthropogenic sulfate (Goto-Azuma and Koerner, 2001). However, the results of our backward-trajectory study do not support this hypothesis, as discussed below.

The 10 d precipitation-weighted backward trajectories for the SIGMA-D and D4 sites showed no contributions of air masses from Antarctica (AT), Australia and New Zealand (AUS), South America (SAM), southeastern Asia (SEA), southern Asia (SA), the Antarctic Ocean (ATO), the South Pacific Ocean (SPO), the Indian Ocean (INO), and the South Atlantic Ocean (SAO). The four regions of the Middle East (ME), Africa (AF), eastern Asia (EA), and central Asia (CA) (Fig. 5) showed maximum contributions of < 0.05 %. In further analyses, we omitted the above 13 regions and focused on the 8 regions of Europe (EU), the Greenland Ice Sheet (GrIS), Russia (RUS), North America (NA), the North Pacific Ocean (NPO), the North Atlantic Ocean (NAO), the Arctic Ocean (AO), and Iceland (IC). Of these, GrIS, AO,

NA, and NAO were found to be the major sources of the air masses arriving at both the SIGMA-D site and the D4 site, although only NA represents a source of anthropogenic BC emissions. Therefore, the temporal trends in anthropogenic rBC at both SIGMA-D and D4 appear to reflect the trend of BC emission in NA.

For precipitation-weighted trajectories, the contributions of air masses from EU and RUS, which are regions with high levels of emission of anthropogenic BC (Hoesly et al., 2018), were less than 4 % and 1 %, respectively, at both the SIGMA-D site and the D4 site, even in winter, when their contributions are at their maxima. To investigate the influence of contributions from EU and RUS in more detail, we recalculated the air mass contributions by excluding GrIS and the oceanic regions of NPO, NAO, and AO where there are no sources of BC emission. Although GrIS had the largest air mass contributions throughout the year and throughout the 10 d, we excluded it because most of the region is covered with ice and has very minor BC sources. The temporal variations in the contributions from NA, EU, RUS, and IC are plotted in Fig. 6 (for both precipitation-weighted and precipitation-unweighted trajectories), and the probability distributions of the air masses for the SIGMA-D and D4 sites are displayed in Fig. 7 (for precipitation-weighted trajectories) and Fig. A5 (for precipitation-unweighted trajectories). Except for the initial few days at D4, contributions from NA are the highest at both the SIGMA-D and D4 sites for both precipitation-weighted and precipitation-unweighted trajectories through-

out the year. The significant contributions of IC during the first few days at the D4 site are likely due to its proximity to Greenland despite its small area, since IC is the only land region near Greenland that can serve as a BC source region. However, if we include oceanic regions along with land regions, the contribution from IC decreases substantially.

At both the SIGMA-D and D4 sites, the contributions from EU and RUS increase in winter (Figs. 6, 7, and A5) when air masses from distant sources can more easily reach the Arctic (Jurányi et al., 2023) for both precipitation-weighted and precipitation-unweighted trajectories. Contrary to our expectations based on CMIP5 emission inventories, the contributions from EU were slightly greater at D4 than at SIGMA-D. Although backward-trajectory analyses showed that contributions from EU were slightly different between the SIGMA-D and D4 sites, the results suggested the opposite conclusion to that of an assumption based on the regional difference in emission inventories to explain the slight differences in the temporal trend of rBC at the two sites. For precipitation-weighted trajectories, the contributions from RUS were similar at both sites and comprised approximately 5 % of the total at most in winter, when the anthropogenic input of rBC is greatest at both the SIGMA-D site (see Sect. 3.2) and the D4 site (McConnell et al., 2007). While precipitation-unweighted trajectories in winter show higher contributions from RUS compared with precipitation-weighted trajectories at both the SIGMA-D and D4 sites, the contribution from RUS is greater at the SIGMA-D site than at the D4 site. The slight difference in temporal patterns of mass concentrations of anthropogenic rBC between the two sites might reflect the different contributions from RUS in winter. However, it should be noted that the 1958–2015 time frame used for back-trajectory calculations, a constraint given by the availability of reanalysis data, is not necessarily representative of the entire 350-year period. Nagatsuka et al. (2021), based on the back-trajectory studies for the SIGMA-D site, have shown that the interannual variability in contributions from different regions remained relatively constant during 1958–2013. Thus, we can discuss the source regions of rBC for this period based on our back-trajectory calculations. For earlier periods, we can only hypothesize the back-trajectories by assuming similar atmospheric circulations to those of the 1958–2015 period. This could lead to large uncertainties for the years before 1958.

Additionally, there are large uncertainties in emission inventories. Although the CMIP5 emission inventories appear to reproduce the temporal patterns in concentrations and fluxes of rBC in Arctic ice cores better than those produced using the Coupled Model Intercomparison Project phase 6 (CMIP6) inventories, the reproduction of the magnitudes of the concentrations and fluxes is better when using the CMIP6 inventories (Eckhardt et al., 2023). A model intercomparison study, which compared the modeling results obtained from 11 Earth system models using CMIP6 emission inventories with rBC records from ice cores (Moseid et al., 2022),

revealed errors in European emission inventories. However, the study also showed that rBC concentrations in northern Greenland ice cores reflected European emissions, contradicting our backward-trajectory analyses. It should also be noted that backward-trajectory analyses are unable to capture the contributions of air masses transported through the upper troposphere (Nagatsuka et al., 2021), which could be important when estimating the contributions from distant sources. Currently, we are unable to explain the slightly different temporal trends in the rBC records from different ice cores in Greenland. Further elucidation of this topic will require additional modeling studies constrained by accurate rBC records from Greenland ice cores.

Figure 8 displays decadal mean mass and number size distributions of rBC for different periods with different anthropogenic inputs. We assumed that the mass size distribution follows a lognormal distribution; thus we estimated the mass median diameter (MMD), which is one of the measures of an rBC size distribution. The decadal mean MMD was 226 nm in the pre-industrial period of 1783–1792. It increased to 325 nm in the peak anthropogenic period of 1913–1922 and subsequently decreased to 302 nm in 1993–2002 and 278 nm in 2003–2012. Number size distributions did not show noticeable temporal change. To investigate the temporal changes in rBC size distribution, we used the average mass of rBC particles (mBC) in addition to the MMD. The parameter mBC can be calculated by dividing the mass concentration by the number concentration. Figure 9 shows the annual and decadal mean mBC and decadal mean MMD together with the annual and decadal mean rBC mass concentrations. Based on breakpoint analyses, we deduced the timing of rBC size changes. The annual mean mBC and decadal mean mBC started to increase in the 1850s and the 1840s, respectively, while the decadal mean MMD started to increase in the 1820s. Annual mean mBC, decadal mean mBC, and decadal mean MMD peaked in the 1920s, when the mass and number concentrations of rBC were at their maxima. The peak values of MMD and mBC were approximately 1.5 and 2 times as high as the corresponding pre-industrial values, respectively. Anthropogenically derived rBC particles that arrived in northwestern Greenland appear to have been larger than rBC particles of biomass burning origin. This is contrary to our expectations because it has been reported that the sizes of rBC particles from biomass burning are larger than those from anthropogenic emissions near the sources (Bond et al., 2013). In the 1920s or 1930s, MMD and mBC both started to decrease, as did the mass and number concentrations of rBC particles. However, in contrast to rBC concentrations, neither MMD nor mBC returned to their pre-industrial levels; instead, they remained approximately 1.3 and 1.5 times higher than their pre-industrial levels, respectively. We also notice that the start of the increases in mBC and MMD appear to have occurred earlier than the increases in mass and number concentrations of rBC by 20–50 years.

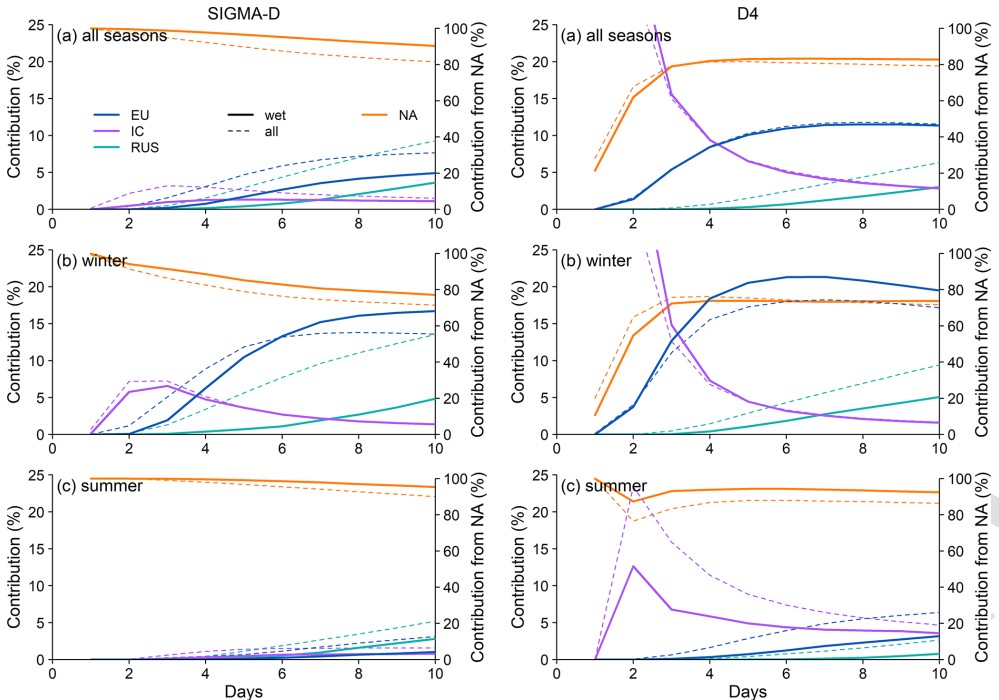

**Figure 6.** Temporal variability in the contribution of air masses arriving at (left) the SIGMA-D site and (right) the D4 site from four regions: **(a)** averages of 12 months, **(b)** averages of winter months (December–February), and **(c)** averages of summer months (May–July). The right-hand axes indicate contributions from NA, and the left-hand axes indicate contributions from the other regions. Solid and dashed curves denote results with and without weighting for the daily local precipitation, respectively.

## 3.2 Temporal changes in seasonal variations in concentrations and sizes of rBC particles

Figure 10 compares the monthly mean rBC mass concentrations in three periods: the pre-industrial period, the period with high anthropogenic input, and recent years when concentrations decreased and returned to pre-industrial levels. Changes are evident in the seasonality of rBC concentrations with respect to Na concentrations, which peak in winter. As reported by McConnell et al. (2007) in relation to the D4 core, rBC concentrations peaked in summer in the pre-industrial period, whereas they peaked in late winter to early spring during the peak industrial period. Figure 10 also indicates that, in recent years after the rBC concentrations returned to their pre-industrial levels, the peaks once again occurred in summer. During the transition periods between the pre-industrial and peak anthropogenic periods and between the peak anthropogenic period and recent years, concentrations show complex seasonal variability. In some years, peaks occurred in both summer and winter/early spring, whereas seasonal peaks were obscured or summer peaks and winter/early spring peaks appeared alternately in other years.

To examine the general temporal trends in seasonal variations in rBC mass concentrations, we plotted 20-year averages of rBC mass concentrations in each month for the years 1653–1992 and we plotted 10-year averages for 1993–2002

(Fig. 11). Up until the 20-year period of 1853–1872, rBC mass concentrations were elevated from March to September, peaking in the late spring to summer months (i.e., May–July). After the 20-year period of 1853–1872, rBC concentrations in autumn to spring increased and became dominant. During the first half of the 20th century, rBC mass concentrations peaked in the winter months (mostly December and January). The autumn-to-spring increases in rBC concentrations are likely attributable to the inflow of anthropogenic emissions (McConnell et al., 2007). The seasonality of the anthropogenic rBC at SIGMA-D is consistent with that of the present-day atmospheric rBC observations at Arctic sites, such as Alert (Canadian High Arctic), Ny-Ålesund (Svalbard), Utqiagvik (Alaska), and a Greenland coastal site (Sharma et al., 2006, 2019; Gong et al., 2010; Qi and Wang, 2019; Massling et al., 2015).

After the 20-year period of 1913–1933, when the anthropogenic input was at its maximum, the autumn-to-spring concentrations decreased. During 1993–2002, the rBC mass concentration peaked in summer again. The recent seasonality of rBC at the SIGMA-D site is the same as that observed at other Greenland ice-coring sites, including EGRIP (Du et al., 2020) and Summit (Fig. 1) (Schmeisser et al., 2018), but it differs from that of atmospheric observations in the Arctic (including Greenland), where rBC concentrations peak in winter/early spring (Sharma et al., 2006, 2019; Gong

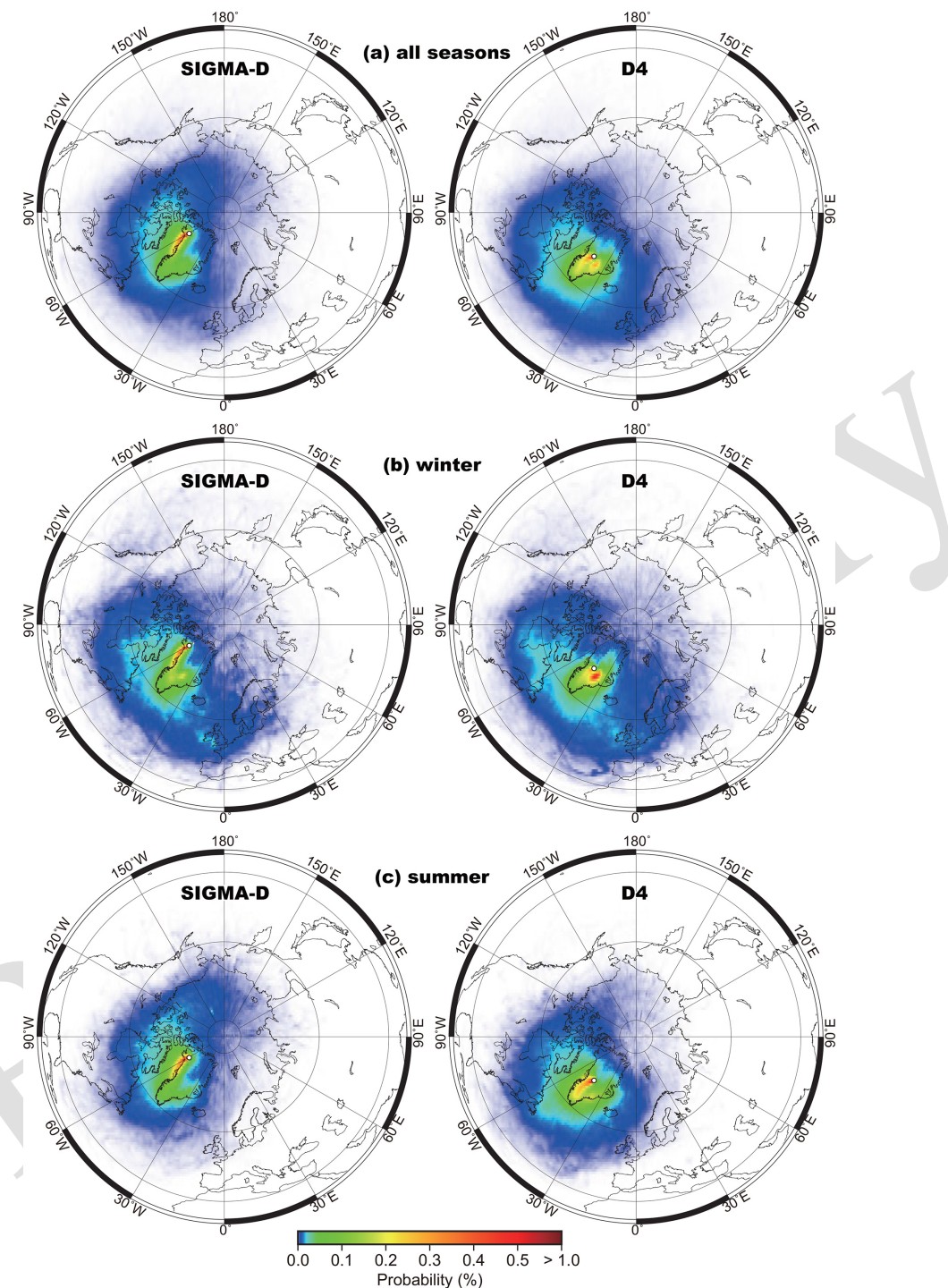

**Figure 7.** Probability distributions of air masses at (left) the SIGMA-D site and (right) the D4 site: **(a)** averages of all seasons, **(b)** averages of winter months (December–February), and **(c)** averages of summer months (May–July).

et al., 2010; Qi and Wang, 2019; Massling et al., 2015). Although we do not present the results for rBC number concentrations, they showed seasonal variations similar to those found in mass concentrations. The influence of anthropogenic emissions in the recent 2 decades appears to be much

lower at the ice-coring sites of SIGMA-D, EGRIP, and Summit, located at elevations of > 2000 m a.s.l., in comparison with that at atmospheric observation sites located near sea level where anthropogenic emissions remain dominant. At the high-elevation sites on the GrIS, concentrations of rBC

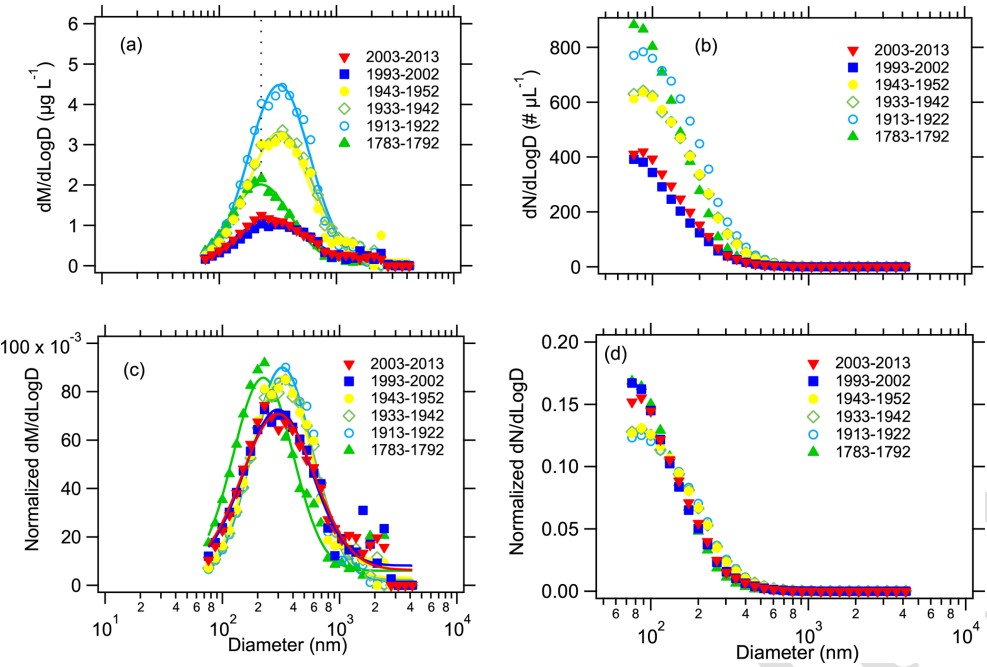

**Figure 8.** Temporal changes in decadal mean size distributions of rBC particles: **(a, b)** non-normalized mass and number size distributions, respectively, and **(c, d)** mass and number size distributions normalized by total rBC mass and number concentrations, respectively. The dotted line in panel **(a)** indicates the mass median diameter (MMD) for the period 1783–1792.

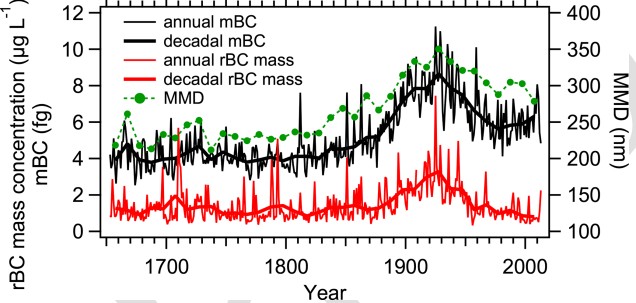

**Figure 9.** Annual and decadal mean mBC (black) and decadal mean MMD (green) together with annual and decadal mean rBC mass concentrations (red). The thin and thick solid curves denote annual and decadal means, respectively.

from biomass burning have exceeded those associated with anthropogenic emissions since the late 20th century, which is likely attributable to reduced emissions of anthropogenically derived rBC, primarily in NA and secondarily in EU (McConnell, 2010; McConnell et al., 2007; Moseid et al., 2022).

To understand the general temporal trends in seasonal variations in rBC size, we plotted 20-year averages of mBC in each month for the years 1653–1992 and we plotted 10-year averages for 1993–2002 (Fig. 11). Up until the 20-year period of 1853–1872, mBC peaked in the spring to summer months (May–July) in most of the 20-year periods and it never peaked in the winter months. After the 20-year period

of 1853–1872, mBC in autumn–spring increased, and its seasonality became obscured. After the peak anthropogenic period of 1913–1933, mBC in autumn–spring decreased. During 1993–2002, mBC once again peaked in summer. We see similar temporal trends in Figs. 12 and 13. Both MMD and mBC showed higher values in summer in the pre-industrial period. This seasonality would indicate that the sizes of rBC particles originated from biomass burning are greater in summer than in winter. The winter and summer mBC started to increase in the 1820s–1830s with a larger rate of increase for winter mBC. Winter and summer values both peaked in the 1890s–1930s and subsequently decreased, with a larger rate of decrease for winter. The winter and summer MMD started to increase in the 1840s and 1810s, respectively, with a larger rate of increase for winter MMD. While winter MMD started to decrease in the 1900s–1920s and has continued to decrease, the summer MMD did not exhibit a clear downward trend. During the peak anthropogenic period, the summer and winter mBC and MMD were close, which obscured the seasonality in rBC particle size (Fig. 11). We also note that, during the peak anthropogenic period, rBC particles larger than 1 μm in diameter increased in winter (Fig. 13). The winter values of MMD and mBC became lower than the summer values in 1993–2002. Larger rBC particles in winter in the anthropogenic period support the argument that rBC particles deposited at SIGMA-D were larger when originating from anthropogenic emissions than when associated with biomass burning.

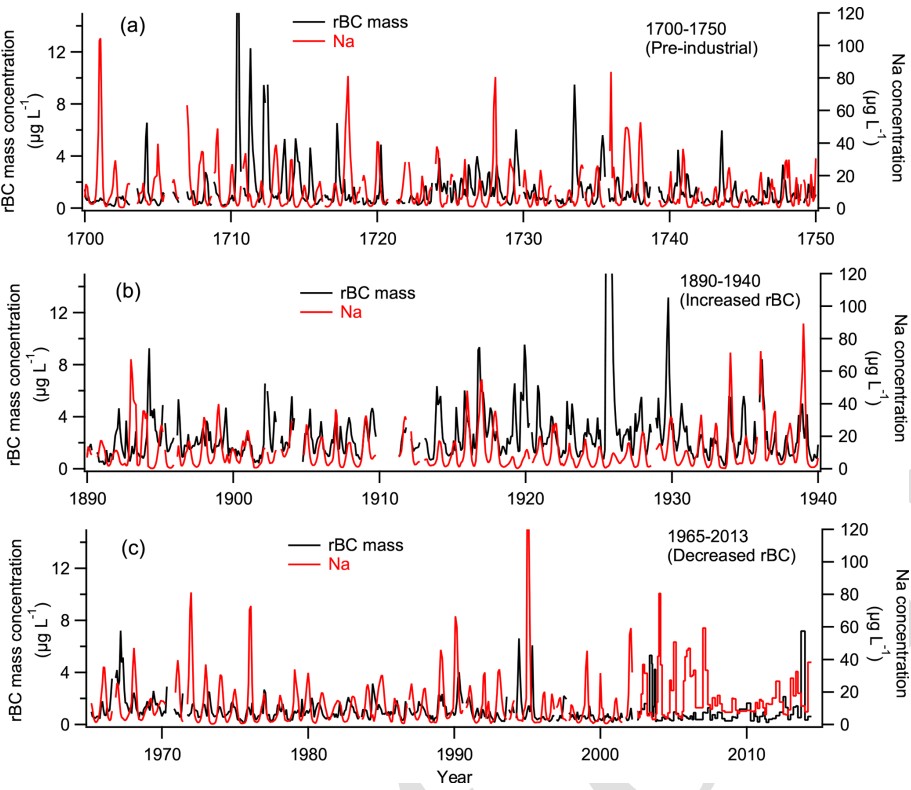

**Figure 10.** Monthly mean rBC mass concentrations and Na concentrations in three periods calculated from the CFA data, with the exception of 2003–2013 (concentrations for this period are raw data from the discrete samples that were analyzed with monthly–bimonthly resolution).

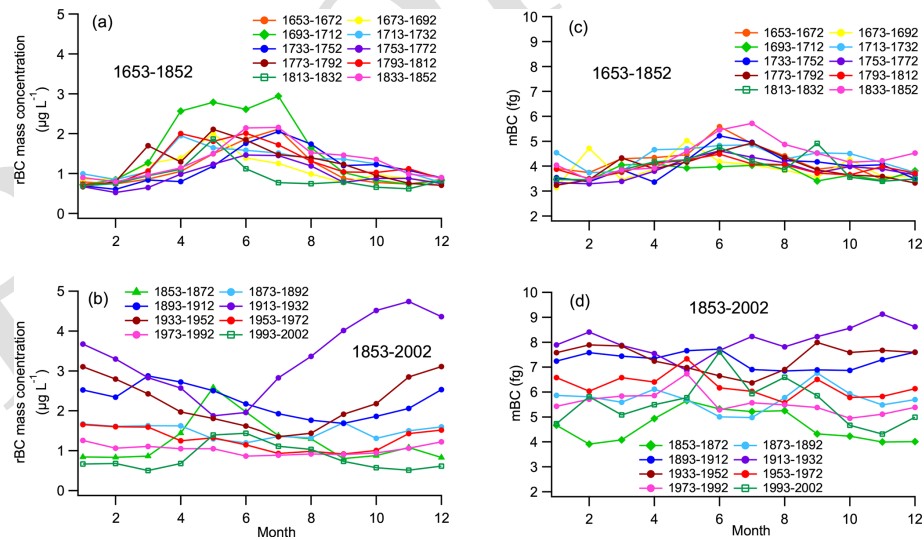

**Figure 11.** The 20-year averages of **(a, b)** rBC mass concentrations and **(c, d)** mBC in each month for the years 1653–1992 and the 10-year averages for 1993–2002.

In the pre-industrial period, biomass burning would have been the predominant source of rBC. Backward-trajectory analyses (Figs. 6, 7, and A5) indicate that boreal forest fires in NA would be the primary sources of rBC in summer at both the SIGMA-D site and the D4 site. Although

the contributions of air masses from RUS are very small, especially in summer (< 3 % at SIGMA-D and < 1 % at D4 for precipitation-weighted trajectories), Siberia has also been proposed as a potential source of pyrogenic aerosols to Greenland (Zennaro et al., 2014). A recent study using

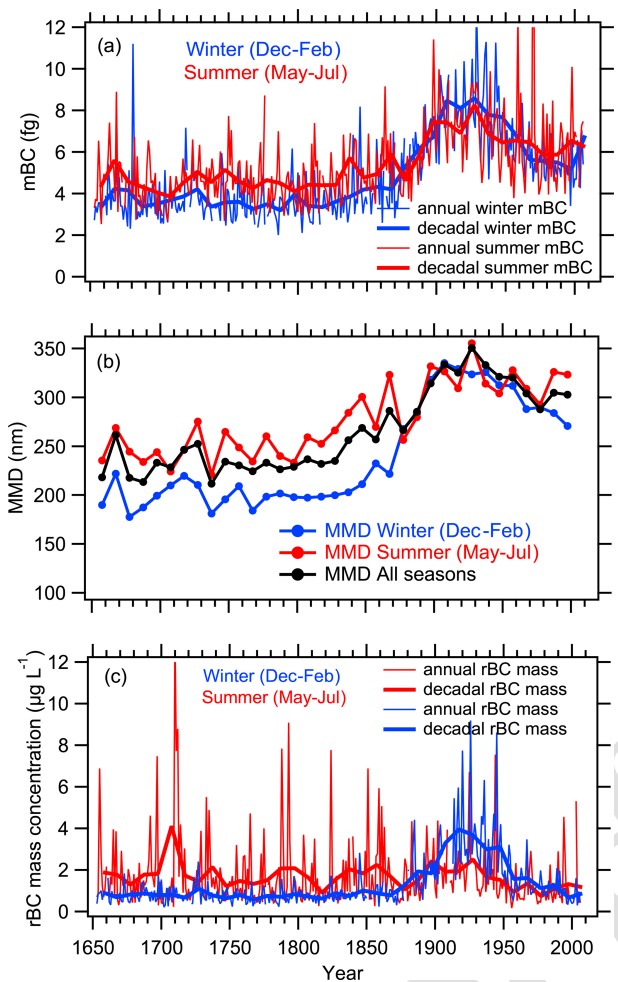

**Figure 12. (a)** Annual and decadal means of winter (December–February) and summer (May–July) mBC; **(b)** decadal means of winter, summer, and all-season (January–December) MMD; and **(c)** annual and decadal means of winter and summer rBC mass concentrations. In all panels, thin and thick solid lines denote annual and decadal means, respectively. Blue, red, and black curves denote winter, summer, and all-season means, respectively.

the CAM-ATRAS global climate–aerosol model (Matsui et al., 2022) showed that Siberia has made the largest contribution to rBC concentrations found in the recent Arctic snow, although the contribution to Greenland snow specifically has not been reported. Therefore, Siberia could be a secondary source of rBC at the SIGMA-D site in summer. In winter, the boreal forests in NA and Siberia are covered with snow; thus there is little contribution of BC from boreal forest fires (Bond et al., 2013). However, rBC concentrations are not zero, even in winter. Biomass burning in lower latitudes (Zennaro et al., 2014) could be a source of rBC in winter, and the smaller sizes of rBC particles in pre-industrial winter periods suggest long-range transport of rBC that supports this assumption.

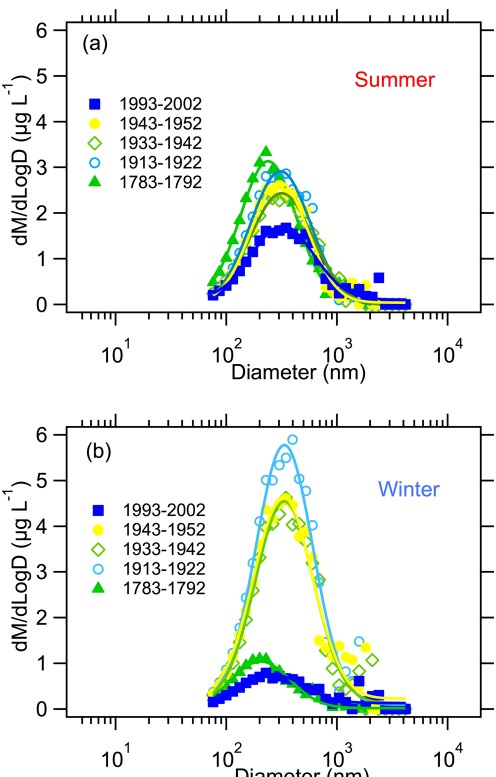

**Figure 13.** Temporal variability in mass size distributions of rBC particles during **(a)** summer (May–July) and **(b)** winter (December–February).

### 3.3 Historical changes in rBC originated from biomass burning

Figure 14 presents monthly mean rBC mass concentrations for the past 350 years, together with $NH_4^+$ concentrations. Occasional high peaks in summer, which lasted for 1–2 months, likely originated from large boreal forest fires, mainly in NA but with possible additional contributions from Siberia. Many of the high rBC peaks in summer coincide with high $NH_4^+$ concentration peaks in summer, which originate from large boreal forest fires (e.g., Legrand et al., 2016). Table A1 lists the biomass burning events distinguished in the record of the SIGMA-D ice core, in other ice cores, and in the surface snow of Greenland. The events distinguished in the SIGMA-D core were defined using rBC mass concentration peaks and $NH_4^+$ concentration peaks. For rBC, peaks exceeding the summer (May–July) averages for 350 years $+ 2\sigma$ or $3\sigma$ were selected, whereas, for $NH_4^+$, summer peaks exceeding the annual averages for 350 years $+ 2\sigma$ or $3\sigma$ were selected. If an rBC or $NH_4^+$ summer peak in the SIGMA-D core with a concentration between the average $+ 1\sigma$ and $2\sigma$ was found in the same year as when a large biomass burning event was recorded at other Greenland sites, we also selected that peak as a biomass burning event. If the year of a biomass burning event reported by previous studies agreed with that in

the SIGMA-D core to within $\pm 2$ years, taking dating errors in different ice cores into account, we assumed that the record in the different cores reflected the same event. In Fig. 14b, we marked only those events exceeding the average $+ 3\sigma$. Most of the marked events (peaks in 1655, 1665, 1697, 1710, 1711, 1712, 1733, 1788, 1793, 1824, 1851, 1859, 1863, 1925, and 1944) occurred in May, June, or July; however, those in 1789, 1812, and 1894 occurred in April; the one in 1773 occurred in September; and the one in 1929 occurred in September–October. The peak in 1925 occurred in July–August, and that in 1944 occurred in May and June, and we assumed that they originated from large biomass burning events. Nevertheless, we could not abandon the possibility that these summer peaks might have been affected by large peaks in the preceding winters owing to signal dispersion in the CFA system.

Most of the large events with rBC concentrations that exceeded the average $+ 3\sigma$ were also recorded in boreal forest fire records reconstructed from rBC, $NH_4^+$, or levoglucosan concentrations in other Greenland ice cores (Table A1). The high rBC concentration peaks in the summers of 1665, 1710, 1711, 1712, and 1824, accompanying high $NH_4^+$ peaks, and the high rBC peaks in 1812 and 1859 have not been reported previously. The high $NH_4^+$ concentration peaks in 1675, 1690, 1740, 1750, and 1756 have no corresponding rBC peaks in April, May, June, July, or August, while those in 1697, 1710, 1712, 1715, and 1761 do have corresponding rBC peaks. Although Keegan et al. (2014) argued that the high summer rBC concentrations in 1889 and 2012 found at Summit were associated with widespread melt events in Greenland, no high rBC concentration peaks were found in 1889 or 2012 at the SIGMA-D site. As shown in Fig. 14 and Table A1, some of the large biomass burning events recorded in other Greenland ice cores were not recorded in the SIGMA-D core and vice versa. There are two possible reasons for this: (1) different ice core sites are not always on the transport pathways of rBC from boreal forest fires (Legrand et al., 2016), and (2) wind scour at a site can remove snow containing high concentrations of rBC. Despite the minor regional differences within Greenland, most of the large rBC concentration peaks caused by large biomass burning events were recorded widely across Greenland. This indicates that high rBC concentration peaks could be used to synchronize different ice cores in Greenland as reference horizons for dating, as is usually carried out with volcanic sulfate peaks and their signatures detected by dielectric profiling (DEP) and electrical conductivity measurement (ECM) peaks (Rasmussen et al., 2008; Sinnl et al., 2022).

The numbers of large biomass burning events in each decade are plotted in Fig. 15 using different definitions of a "large" event. Figure 15a displays the number of months with mass concentrations in summer (May–July) exceeding the summer average $+ 1\sigma$, $2\sigma$, and $3\sigma$; Fig. 15b displays the number of months with number concentrations in summer exceeding the summer average $+ 1\sigma$, $2\sigma$, and $3\sigma$. Because large events in April, August, and September were not

counted to avoid the potential for impact of anthropogenic rBC, there would be minor underestimation of the number of large events. Although the frequency of large events differs slightly between the different definitions, the general tendency is consistent. Large events tended to be more frequent around the 1710s, 1790s, 1850s, 1900s, and 1950s. Moreover, there is no obvious trend in increase up to the decade 1993–2002. To study the historical trends in concentrations of rBC originated from biomass burning, we calculated the decadal averages of rBC mass and number concentrations for each month (Fig. 16). During the pre-industrial period, both mass and number concentrations were stable in the winter months (December–February), whereas they showed large inter-decadal fluctuations in the spring to autumn months (March–November). The fluctuations appear largest in spring–summer (April–July). Generally, the period of April–July is likely when the occurrence of large boreal forest fires increases in NA (Whitman et al., 2018). The sporadic nature of the frequency of occurrence of large boreal forest fires would explain the large fluctuations.

Since the 1870s, when anthropogenic rBC started to influence the SIGMA-D site, rBC mass and number concentrations in September–April increased, as discussed in Sect. 3.2; however, there was little increase during the spring–summer months. The same feature is also seen in Fig. 12c. Although large inter-decadal variability in concentrations during spring–summer obscured the temporal trends in spring–summer concentrations (Fig. 11), the general temporal trends are more apparent in Fig. 16. At the SIGMA-D site, we see slight trends in reduction in rBC mass and number concentrations during spring–summer. Analysis of Fig. 15 also suggests that the frequency of large boreal forest fires in NA showed a slight trend of reduction over the past 350 years until the most recent decade (1993–2002). However, this trend has not been reported by previous studies on other ice cores from Greenland, partly owing to the different periods covered, the different temporal resolution of the analysis methods, and the different fire proxies used (Zennaro et al., 2014; Legrand et al., 2016; Parvin et al., 2019; Savarino and Legrand, 1998; Whitlow et al., 1994). Most previous studies used $NH_4^+$ as a fire proxy, with the occasional use of levoglucosan and other organic materials; only a few studies have used rBC as a fire proxy for pre-industrial periods (Zennaro et al., 2014).

Since the 1950s, data on the area burned and the number of forest fires in Canada have become available (Hanes et al., 2018; Skakun et al., 2021). The SIGMA-D record does not appear to trace the Canadian forest fire database. Air masses arriving at the SIGMA-D site might not have passed over Canada during periods of large forest fires. Large uncertainties in fire data might also explain the disagreement. There are also large uncertainties and regional variability in sedimentary charcoal fire records (Marlon et al., 2012, 2013; Power et al., 2013). However, our results are consistent with the charcoal data from western NA, which show a

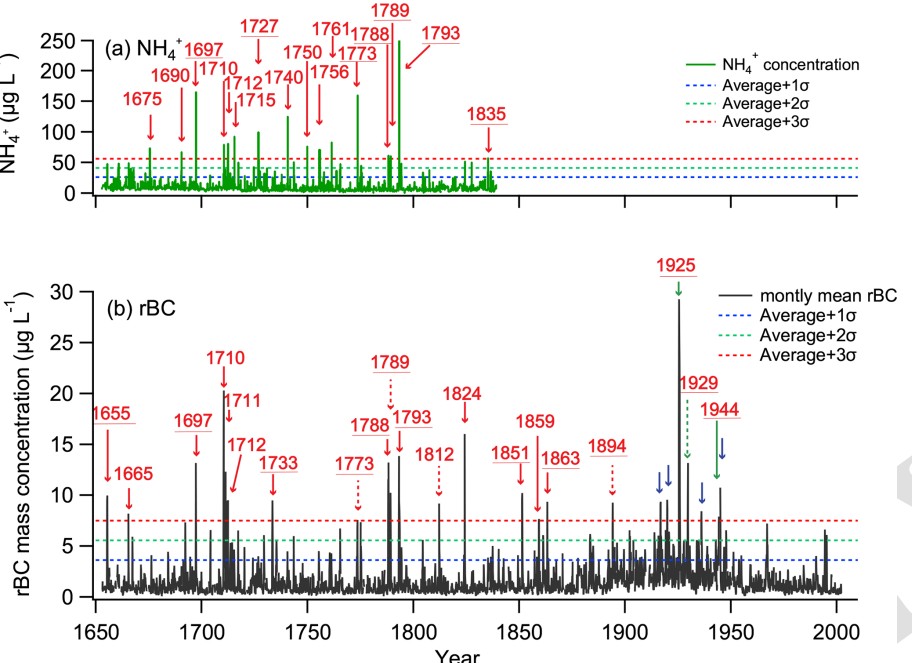

**Figure 14. (a)** $NH_4^+$ concentrations (raw data) and **(b)** monthly mean rBC mass concentrations calculated from CFA data. Red arrows in panel **(a)** show years when concentrations exceeded the average $+3\sigma$ in summer. Solid and broken arrows in panel **(b)** show years when concentrations exceeded the summer (May–July) average $+3\sigma$ in summer (May–July) and in spring or autumn months (April, August, and September), respectively. Blue arrows in panel **(b)** show years when concentrations exceeded the summer average $+3\sigma$ in winter. Solid and broken green arrows in panel **(b)** show years when concentrations exceeded the summer (May–July) average $+3\sigma$ in summer (May–July) and in spring or autumn months (April, August, and September), respectively, but those that might have been affected by winter peaks. Underlining denotes years when biomass burning events were also recorded in other Greenland ice cores/surface snow samples within 2 years. Dotted lines in panel **(a)** denote the concentration average over 350 years $+1\sigma$ (blue), $+2\sigma$ (green), and $+3\sigma$ (red). Dotted lines in panel **(b)** denote the summer (May–July) concentration average over 350 years $+1\sigma$ (blue), $+2\sigma$ (green), and $+3\sigma$ (red).

general decline in biomass burning since 1500, with a relatively enhanced fire period in the mid-19th century (Power et al., 2013). The biomass burning emission inventories for CMIP6 also have large uncertainties (van Marle et al., 2017). Therefore, much more work is needed on the reconstruction of past biomass burning using ice cores.

## 3.4 Impacts of rBC on the ice sheet albedo

McConnell et al. (2007) calculated the quantitative impacts of rBC in snow on radiative forcing during early summer using rBC concentration from the D4 ice core and the Snow, Ice, and Aerosol Radiative (SNICAR) model (Flanner et al., 2007), assuming an effective snow grain radius of $100\,\mu m$. They estimated the radiative forcing of $1.02\,W\,m^{-2}$ during the peak 5-year period from 1906 to 1910, representing a 5-fold increase compared to pre-industrial conditions. However, the radiative forcing of rBC in snow varies depending on both the assumed solar irradiance and the snow grain size. In this study, we calculated the possible albedo reduction due to rBC at the SIGMA-D site from the monthly mean rBC mass concentration data obtained in this study (Fig. 17) using a physically based snow albedo model (Aoki

et al., 2011). As the snow albedo reduction rate due to light-absorbing particles is enhanced with an increase in snow grain size (Wiscombe and Warren, 1980), we assumed two effective snow grain radii, $r_s = 50$ and $1000\,\mu m$, corresponding to new snow (defined as "precipitation particles" according to Fierz et al., 2009) and old melting snow (defined as "melt forms" according to Fierz et al., 2009) (Wiscombe and Warren, 1980) for clear sky and cloudy sky (overcast) conditions. The albedo reductions under the cloudy sky are 20 %–48 % larger than those under the clear sky. These increases are related to the following two factors: (1) the visible albedo depends more strongly on rBC concentration than the near-infrared albedo (Wiscombe and Warren, 1980); (2) the visible spectral fraction in solar irradiance at the snow surface under cloudy sky is larger than that under clear sky (Aoki et al., 1999). Thus, the albedo reduction due to rBC under cloudy sky is enhanced more than under clear sky.

Figure 17 demonstrates that the albedo reduction in the case of new snow is consistently less than 0.01, even at the maximum value for cloudy conditions in August 1925. In contrast, for old melting snow, the albedo reduction frequently exceeds 0.01 for both sky conditions. The maximum albedo reduction for cloudy conditions is 0.045

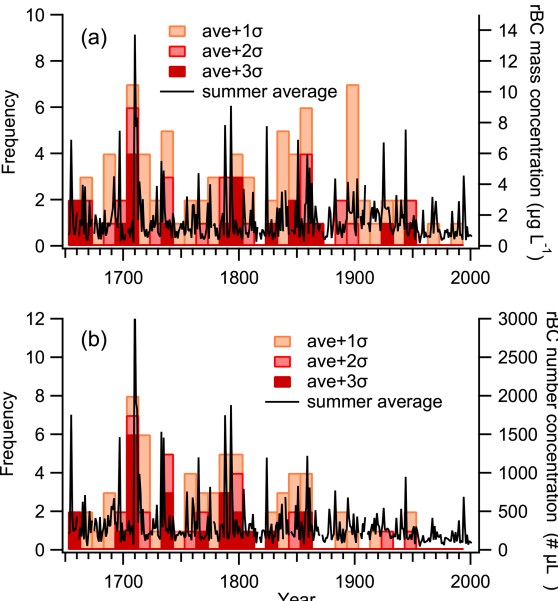

**Figure 15.** Frequency of large biomass burning events. Black curves denote summer (May–July) averages of rBC concentrations. Bars show the frequency of rBC summer peaks in each decade exceeding the average $+ 1\sigma$, $2\sigma$, and $3\sigma$. **(a)** Mass concentrations of rBC were used to define peaks. **(b)** Number concentrations of rBC were used to define peaks.

($r_s = 1000\,\mu m$) and 0.0098 ($r_s = 50\,\mu m$) in August 1925, followed by the values of 0.034 ($r_s = 1000\,\mu m$) and 0.0074 ($r_s = 50\,\mu m$) in July 1710. The averaged albedo reduction for the overall period of 1650–2014 is 0.0031 ($r_s = 1000\,\mu m$) and 0.0007 ($r_s = 50\,\mu m$) for clear conditions and 0.0054 ($r_s = 1000\,\mu m$) and 0.0011 ($r_s = 50\,\mu m$) for cloudy conditions. During the anthropogenic concentration peak period of 1913–1933, the average albedo reduction increases to 0.0056 ($r_s = 1000\,\mu m$) and 0.0012 ($r_s = 50\,\mu m$) for clear conditions and to 0.0089 ($r_s = 1000\,\mu m$) and 0.0019 ($r_s = 50\,\mu m$) for cloudy conditions.

Warren (2019) described that an rBC concentration of 20 parts per billion in weight can cause broadband snow albedo reductions of 1 %–2 %. He also noted that, for a typical daily average solar irradiance of $400\,W\,m^{-2}$ in the Arctic during late spring and early summer, a 1 % albedo reduction can lead to a positive forcing of $4\,W\,m^{-2}$ locally, similar to the forcing caused by doubling $CO_2$. Our calculation results indicate that a 1 % reduction in albedo can occur at numerous local spike-like peaks for $r_s = 1000\,\mu m$, including the recent several decades after 1950. During the anthropogenic concentration peak period (1913–1933), the average albedo reduction approaches 1 % (0.0089) for $r_s = 1000\,\mu m$ under cloudy sky conditions. Consequently, our simulations suggest that the amount of albedo reduction remains relatively small as long as new snow conditions are maintained. However, if the surface snow grains reach the size of old melting

snow, which would have occurred during summer months at the SIGMA-D site, the extent of albedo reduction becomes non-negligible.

## 4 Conclusions

We analyzed the record of rBC over the past 350 years in the SIGMA-D ice core, which was drilled in northwestern Greenland. The improved technique for rBC measurement (Mori et al., 2016) and the CFA system built at NIPR allowed us to reconstruct high-temporal-resolution records of the sizes and concentrations of rBC particles. Notably, this study marks the first reconstruction of temporal changes in rBC size since the pre-industrial period.

The number and mass concentrations of rBC started to increase in the 1870s owing to anthropogenic input. The concentrations reached their maxima in the 1910s–1920s, following which they decreased. By the 1960s, rBC concentrations had reduced to levels close to or lower than those of the pre-industrial period. The trend in anthropogenic rBC at the SIGMA-D site was generally similar to that reported previously for other ice core sites in Greenland, albeit with slight differences. Backward-trajectory analyses suggest that the major anthropogenic emission source that affected the SIGMA-D site was NA. However, the backward-trajectory analyses did not clearly explain the slight difference in the temporal trends in rBC between the SIGMA-D site and the more southerly site D4. Anthropogenic rBC was transported to the SIGMA-D site mainly in the winter half of the year, which was deduced by the changes in the seasonality of rBC concentrations. The backward-trajectory analyses produced consistent results, showing greater contributions from air masses from the industrial regions in NA, EU, and RUS in winter.

Pre-industrial rBC concentrations peaked in summer. In association with increased anthropogenic input, concentrations increased in winter to early spring, which shifted the annual peak in concentration to winter–early spring. When the anthropogenic input started to decline in the 1930s, concentrations in winter–early spring also decreased, which changed the seasonality of rBC concentrations; by the 1990s, rBC concentrations peaked in summer once again. This suggests that the major sources of rBC in recent years were not anthropogenic emissions but biomass burning. At the SIGMA-D site, rBC originating from anthropogenic emissions made only a minor contribution to the rBC concentrations in the summer months throughout the past 350 years. This enabled us to examine the temporal variability in biomass burning throughout the past 350 years, especially after the increase in anthropogenically derived rBC, which is a topic that has not been addressed by previous studies on rBC data from other ice cores in the Arctic.

The size distributions of rBC particles have also changed owing to anthropogenic impact. The seasonality of MMD

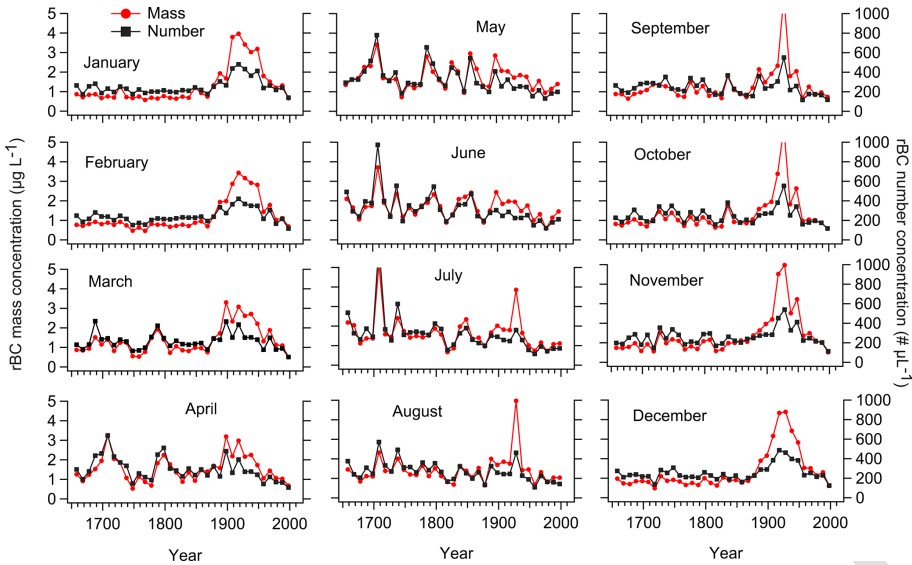

**Figure 16.** Decadal averages of rBC mass (red) and number (black) concentrations for each month.

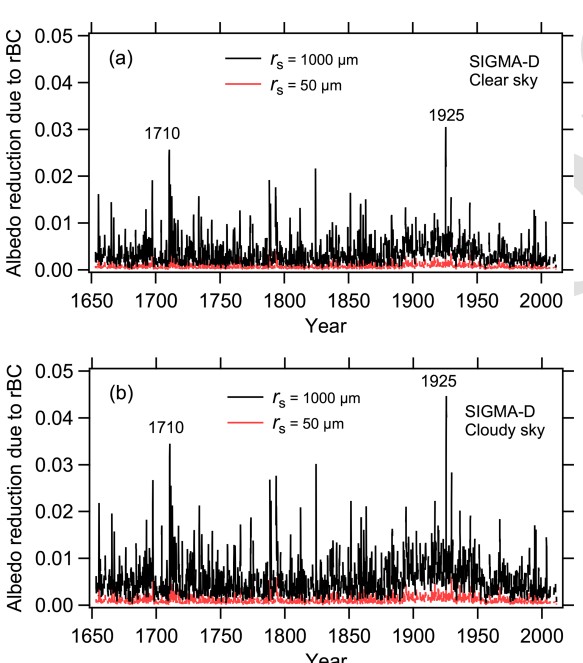

**Figure 17.** Temporal variation in albedo reduction for **(a)** clear sky and **(b)** cloudy sky conditions, calculated from monthly mean rBC mass concentrations, for two effective snow grain radii: $r_s = 50\,\mu m$ (representing new snow, indicated by red curves) and $r_s = 1000\,\mu m$ (indicating old melting snow, shown by black curves). The albedo reduction values are calculated for the solar zenith angle at local solar noon on the 15th of each month, excluding the polar night period from November to February. The maximum albedo reduction is simulated in 1925 and followed by 1710.

and mBC also changed, associated with the concentration changes. Winter and summer mBC started to increase in the 1820s–1830s, peaked in the 1890s–1930s when rBC concentrations peaked, and started to decline since the 1930s. The winter and summer MMD started to increase in the 1840s and 1810s, respectively, with larger rate of increase for winter MMD. While winter MMD started to decrease in the 1900s–1920s and has continued to decrease, the summer MMD did not exhibit a clear downward trend. Increases in winter MMD and mBC associated with the increase in winter rBC concentrations suggest that the diameter of rBC particles deposited over northwestern Greenland were generally larger for anthropogenic rBC than for biomass burning rBC. In contrast to rBC concentrations, neither MMD nor mBC returned to their pre-industrial values; instead, they remained at higher values in the 1960s–2000s.

Pre-industrial rBC would have originated mainly from biomass burning. During the winters (December–February) of the pre-industrial period, decadal averages of monthly mean mass and number concentrations were stable, and the sizes of rBC particles were smaller than those in summer. This indicates that rBC in pre-industrial winters originated from biomass burning in low latitudes, where there was no snow cover in winter, and that biomass burning in low latitudes that affected Greenland showed little change during the pre-industrial period. After the inflow of anthropogenic rBC started, it became difficult to distinguish biomass burning rBC from anthropogenic rBC in winter, making it difficult to discuss the temporal changes in rBC originating from low-latitude biomass burning in winter. However, we could discuss the temporal changes in boreal forest fires that occur mainly in summer, the season with minimal anthropogenic input.

Sources of pre-industrial summer rBC were likely boreal forest fires primarily in NA. We investigated the temporal trend in the decadal frequency of large boreal forest fire events using high summer peaks of number and mass concentrations of rBC. We found no obvious trend in increase in the decadal frequency of large boreal forest fires until the decade of 1993–2002. Furthermore, we found no trends in increase in the decadal averages of monthly mean mass and number concentrations in summer during the past 350 years; we even found a trend in decrease for number concentrations. Although recent large fire events in NA are attributed to global warming (Brown et al., 2023), the effects of global warming do not seem to have left a clear imprint in Greenland until the early 2000s. Therefore, we need further investigations using more recent ice core records of rBC.

We analyzed the temporal variation in potential albedo reduction due to rBC at the SIGMA-D site during the past 350 years using a physically based snow albedo model. Albedo reductions under the assumption of new snow grain size remained below 0.01, even at the peak rBC concentration in 1925. Conversely, under the assumption of old melting snow grain size, the albedo reduction frequently exceeded 0.01. Our calculation results reveal that a 1 % reduction in albedo can occur at numerous local spike-like peaks in the case of old melting snow, including the recent several decades after 1950. During the peak period of anthropogenic concentrations (1913–1933), the averaged albedo reduction approaches 1 % for old melting snow cases. Consequently, our simulations suggest that the magnitude of albedo reduction remains relatively small as long as new snow conditions are maintained. However, if the surface snow grains reach the size of old melting snow, the amount of albedo reduction becomes non-negligible.

Our new high-temporal-resolution records of rBC concentrations and sizes could contribute to the evaluation of the impacts of both anthropogenically derived rBC and rBC originating from biomass burning on Earth's radiation budget, albedo, rBC–cloud interactions, and therefore rBC–climate interactions. They could also contribute to the validation of emission inventories and of aerosol and climate models. High-temporal-resolution rBC data since 2002 are necessary to investigate the impact of global warming on boreal forest fires. Furthermore, high-temporal-resolution records of rBC concentrations and sizes during the early Holocene and the Last Interglacial, when it was warmer than the present day (NEEM community members, 2013; Vinther et al., 2009), should be obtained for better projections of rBC–climate interactions in a future warming world.

## Appendix A

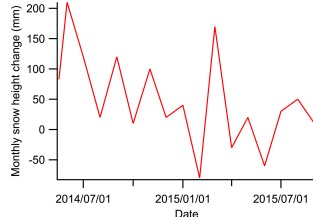

**Figure A1.** Monthly snow height change observed with an automatic weather station (AWS) at the SIGMA-D site during the period May 2014–September 2015 (Matoba et al., 2015). It appears that there was more precipitation in summer than in winter, which could introduce some bias in monthly dating of the SIGMA-D ice core. Although the AWS data indicated that precipitation occurred in all months (Fig. A2), there were a few months when snow height change was negative, mainly owing to wind scouring. Moreover, the seasonal variation in precipitation seems to exhibit significant year-to-year variability. However, by averaging monthly mean concentrations over 10–20 years (Fig. 11), we can observe changes in the seasonality of rBC.

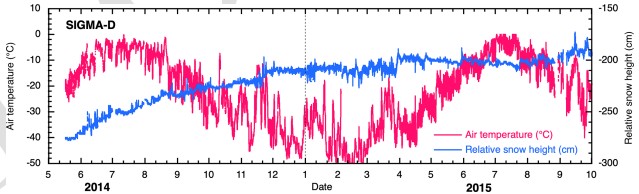

**Figure A2.** Seasonal variations in air temperature and relative snow height every 10 min observed at the SIGMA-D site during the period May 2014–September 2015 (Matoba et al., 2015).

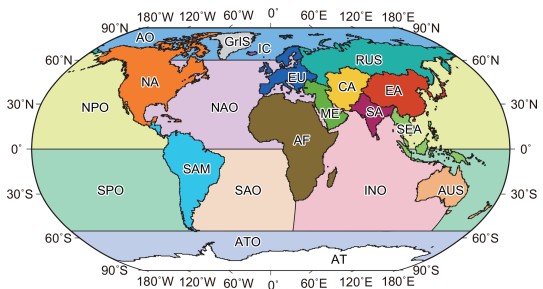

**Figure A3.** Map showing the 21 regions used for calculating the regional contributions in backward-trajectory analysis. ME: Middle East. AT: Antarctica. AUS: Australia and New Zealand. SAM: South America. AF: Africa. EA: China and eastern Asia. EU: Europe. GrIS: Greenland Ice Sheet. SEA: southeastern Asia. SA: southern Asia. CA: central Asia. RUS: Russia. NA: North America. ATO: Antarctic Ocean. SPO: South Pacific Ocean. INO: Indian Ocean. NPO: North Pacific Ocean. SAO: South Atlantic Ocean. NAO: North Atlantic Ocean. AO: Arctic Ocean. IC: Iceland.

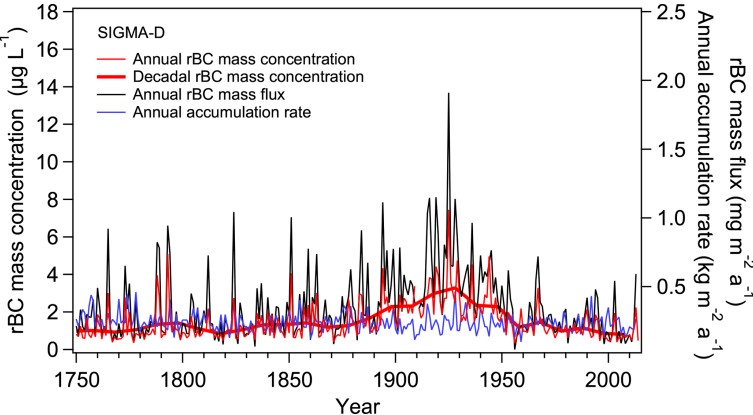

**Figure A4.** Annual mean and decadal mean rBC mass concentrations (thin and thick red curves, respectively), annual rBC mass flux (black), and annual accumulation rate (blue).

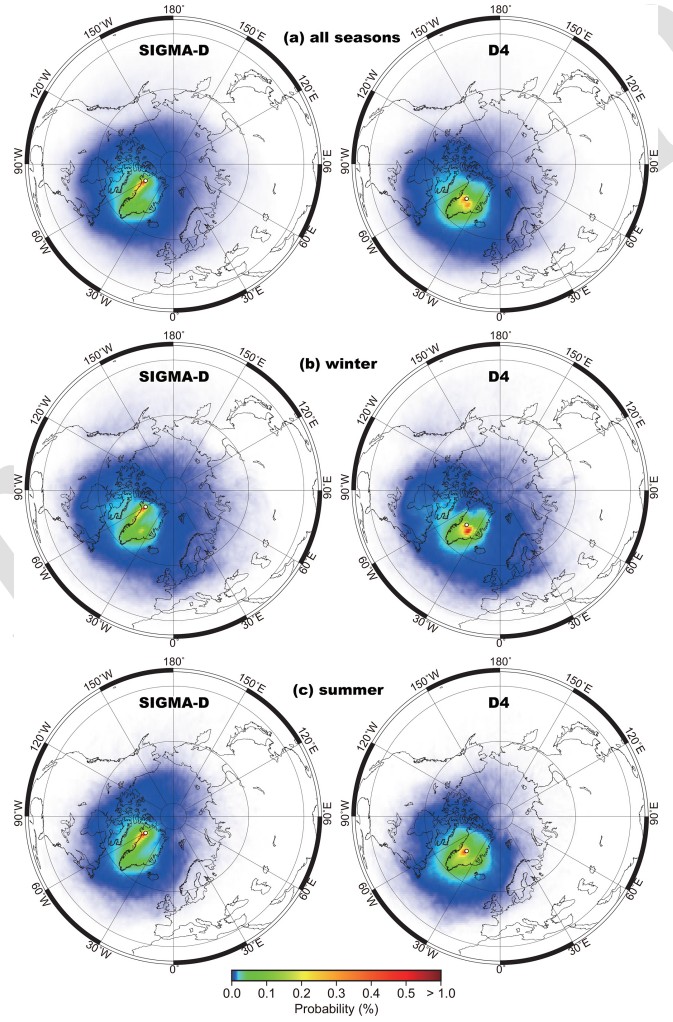

**Figure A5.** Probability distributions of air masses at (left) the SIGMA-D site and (right) the D4 site calculated without weighting with the local daily precipitation: **(a)** averages of all seasons, **(b)** averages of winter months (December–February), and **(c)** averages of summer months (May–July).

**Table A1.** Biomass burning events distinguished in the SIGMA-D ice core record and in the records of other ice cores and surface snow from Greenland. Records from Greenland sites with a temporal resolution of greater than 1 year were not used. Events distinguished in the SIGMA-D core were defined using rBC mass concentration peaks and $NH_4^+$ concentration peaks. For rBC, peaks exceeding the summer (May–July) averages $+2\sigma$ or $3\sigma$ were selected; for $NH_4^+$, peaks exceeding the annual average $+2\sigma$ or $3\sigma$ were selected. If an rBC peak was found in April, August, or September, the year is written in parentheses. If an rBC or $NH_4^+$ summer peak in the SIGMA-D core with a concentration between the average $+1\sigma$ and $2\sigma$ was found in the same year when a large biomass burning event was recorded at other Greenland sites, the peak was also selected as a biomass burning event. If a year with a biomass burning event reported in previous studies agreed with that in the SIGMA-D core within 2 years, the records were considered to reflect the same event and are written on the same line in the table. A: Year of the event in the SIGMA-D core. B: Magnitude of the rBC mass concentration peak. C: Magnitude of the $NH_4^+$ concentration peak. D: Year of the event in ice cores and surface snow from Greenland sites other than the SIGMA-D site. E: Location of the ice core or surface snow in Greenland. F: Biomass burning proxies. G: References.

| SIGMA-D core (this study) | | | Ice cores and surface snow from Greenland sites other than the SIGMA-D site | | | |
|---|---|---|---|---|---|---|
| A | B | C | D | E | F | G |
| Year | Magnitude of rBC peak | Magnitude of $NH_4^+$ peak | Year | Location of ice core or surface snow | Proxy | Reference |
| 1655 | > average + 3σ | > average + 2σ | 1654 | SUMMIT, NEEM | $NH_4^+$ | Legrand et al. (2016) |
|  |  |  | 1655 | SUMMIT | $NH_4^+$, HCOO⁻ | Savarino and Legrand (1998) |
| 1661 | No data | > average + 2σ | | | | |
| 1665 | > average + 3σ | > average + 2σ | | | | |
| 1667 | > average + 2σ | > average + 1σ | | | | |
| 1675 | | > average + 3σ | | | | |
| 1684 | > average + 1σ | | 1683 | NEEM | $NH_4^+$ | Legrand et al. (2016) |
| 1690 | | > average + 3σ | | | | |
| 1692 | > average + 2σ | | | | | |
| 1697 | > average + 3σ | > average + 3σ | 1699 | SUMMIT | $NH_4^+$ | Legrand et al. (2016) |
|  |  |  | 1702–03 | NEEM | $NH_4^+$ | Legrand et al. (2016) |
|  |  |  | 1702 | NEEM | rBC | Zennaro et al. (2014) |
|  |  |  | 1703 | NEEM | rBC | Zennaro et al. (2014) |
| 1710 | > average + 3σ | > average + 3σ | | | | |
| 1711 | > average + 3σ | > average + 1σ | | | | |
| 1712 | > average + 3σ | > average + 3σ | | | | |
| 1715 | > average + 1σ | > average + 3σ | | | | |
| 1717 | > average + 1σ | > average + 2σ | 1719 | NEEM | $NH_4^+$ | Legrand et al. (2016) |
| 1724 | | > average + 2σ | | | | |
| 1727 | | > average + 3σ | 1728 | SUMMIT | $NH_4^+$ | Legrand et al. (2016) |
| 1729 | > average + 2σ | | | | | |
| 1733 | > average + 3σ | | 1732 | NEEM | $NH_4^+$ | Legrand et al. (2016) |
| 1735 | > average + 2σ | | | | | |
| 1740 | | > average + 3σ | | | | |
| 1743 | > average + 1σ | > average + 2σ | | | | |
| 1750 | | > average + 3σ | | | | |
| 1756 | | > average + 3σ | | | | |
| 1761 | > average + 1σ | > average + 3σ | | | | |
| 1765 | > average + 2σ | > average + 2σ | | | | |
| 1773 | > average + 1σ (> average + 3σ in September) | > average + 3σ | 1771 | NEEM | $NH_4^+$ | Legrand et al. (2016) |
|  |  |  | 1773 | SUMMIT | $NH_4^+$ | Legrand et al. (2016) |
|  |  |  | 1773 | SUMMIT | $NH_4^+$, HCOO⁻ | Savarino and Legrand (1998) |
| 1775 | > average + 1σ | > average + 2σ | | | | |
| 1788 | > average + 3σ | > average + 3σ | 1789 | NEEM | rBC | Zennaro et al., 2014) |
| 1789 | > average + 2σ (> average + 3σ in April) | > average + 3σ | | | | |

| | SIGMA-D core (this study) | | | Ice cores and surface snow from Greenland sites other than the SIGMA-D site | | |
|---|---|---|---|---|---|---|
| A | B | C | D | E | F | G |
| Year | Magnitude of rBC peak | Magnitude of $NH_4^+$ peak | Year | Location of ice core or surface snow | Proxy | Reference |
| 1793 | > average $+\,3\sigma$ | > average $+\,3\sigma$ | 1792 | NEEM | $NH_4^+$ | Legrand et al. (2016) |
| 1794 | > average $+\,1\sigma$ | > average $+\,2\sigma$ | 1794–95 | SUMMIT | $NH_4^+$ | Legrand et al. (2016) |
| | | | 1795 | SUMMIT | $NH_4^+$, $HCOO^-$ | Legrand and De Angelis (1996) |
| | | | 1795 | D4 | $NH_4^+$ | Legrand et al. (2016) |
| 1804 | > average $+\,2\sigma$ | > average $+\,1\sigma$ | | | | |
| 1807 | | > average $+\,1\sigma$ | 1807 | SUMMIT, NEEM, D4 | $NH_4^+$ | Legrand et al. (2016) |
| (1812) | (> average $+\,3\sigma$ in April) | | | | | |
| 1824 | > average $+\,3\sigma$ | > average $+\,2\sigma$ | | | | |
| 1827 | | > average $+\,2\sigma$ | | | | |
| 1835 | > average $+\,1\sigma$ | > average $+\,3\sigma$ | 1837 | SUMMIT, NEEM, D4 | $NH_4^+$ | Legrand et al. (2016) |
| 1840 | > average $+\,1\sigma$ | No data | 1839 | SUMMIT | $NH_4^+$ | Legrand et al. (2016) |
| | | | | SUMMIT | $NH_4^+$, HCOO | Legrand and De Angelis (1996) |
| | | | 1846 | SUMMIT | $NH_4^+$ | Legrand et al. (2016) |
| 1849 | > average $+\,1\sigma$ | No data | 1847 | D4 | $NH_4^+$ | Legrand et al. (2016) |
| | | | 1848 | NEEM | $NH_4^+$ | Legrand et al. (2016) |
| 1851 | > average $+\,3\sigma$ | No data | 1853 | NEEM | $NH_4^+$ | Legrand et al. (2016) |
| | | No data | 1854 | SUMMIT, D4 | $NH_4^+$ | Legrand et al. (2016) |
| 1859 | > average $+\,3\sigma$ | | | | | |
| 1861 | > average $+\,2\sigma$ | > average $+\,1\sigma$ | | | | |
| 1863 | > average $+\,3\sigma$ | No data | 1863 | SUMMIT, NEEM, D4 | $NH_4^+$ | Legrand et al. (2016) |
| | | | | SUMMIT | $NH_4^+$, $HCOO^-$ | Savarino and Legrand (1998) |
| | | No data | 1868 | SUMMIT | $NH_4^+$ | Keegan et al. (2014) |
| | | | 1869 | D4 | $NH_4^+$ | Legrand et al. (2016) |
| | No data | No data | 1871 | NEEM, D4 | $NH_4^+$ | Legrand et al. (2016) |
| | | | 1872 | SUMMIT | $NH_4^+$ | Legrand et al. (2016) |
| 1883 | > average $+\,2\sigma$ | No data | 1885 | GISP2 (SUMMIT) | $NH_4^+$ | Whitlow et al. (1994) |
| | | No data | 1886 | SUMMIT, NEEM, D4 | $NH_4^+$ | Legrand et al. (2016) |
| | | | | 20D (Dye-3) | $NH_4^+$ | Whitlow et al. (1994) |
| | | | 1888 | NEEM | $NH_4^+$ | Legrand et al. (2016) |
| | | No data | 1889 | SUMMIT | rBC, $NH_4^+$ | Keegan et al. (2014) |
| | | | | D4 | $NH_4^+$ | Legrand et al. (2016) |
| | | | 1890 | SUMMIT | $NH_4^+$ | Legrand et al. (2016) |
| 1894 | > average $+\,1\sigma$ (> average $+\,3\sigma$ in April) | No data | 1894–95 | SUMMIT, NEEM, D4 | $NH_4^+$ | Legrand et al. (2016) |
| | | | 1895 | SUMMIT | $NH_4^+$, $HCOO^-$ | Legrand and De Angelis (1996) |
| 1896 | > average $+\,1\sigma$ | No data | 1896 | SUMMIT | $NH_4^+$, $HCOO^-$ | Savarino and Legrand (1998) |
| 1902 | > average $+\,2\sigma$ | No data | | | | |
| 1909 | > average $+\,1\sigma$ | No data | 1908 | SUMMIT, D4 | $NH_4^+$ | Legrand et al. (2016) |
| | | | | SUMMIT | $NH_4^+$, $HCOO^-$ | Savarino and Legrand (1998) |
| | | | | SUMMIT | rBC, $NH_4^+$ | Keegan et al. (2014) |
| | | No data | 1921 | NEEM | $NH_4^+$ | Legrand et al. (2016) |
| 1925 | > average $+\,3\sigma$ | No data | 1923 | SUMMIT, D4 | $NH_4^+$ | Legrand et al. (2016) |
| 1927 | > average $+\,1\sigma$ | No data | 1927–28 | NEEM | $NH_4^+$ | Legrand et al. (2016) |
| (1929) | (> average $+\,3\sigma$ in September and October) | | 1929 | SUMMIT, D4 | $NH_4^+$ | Legrand et al. (2016) |

| SIGMA-D core (this study) | | | Ice cores and surface snow from Greenland sites other than the SIGMA-D site | | | |
|---|---|---|---|---|---|---|
| A | B | C | D | E | F | G |
| Year | Magnitude of rBC peak | Magnitude of $NH_4^+$ peak | Year | Location of ice core or surface snow | Proxy | Reference |
| 1936 | > average $+ 1\sigma$ | No data | 1936–38 | SUMMIT | $NH_4^+$ | Legrand et al. (2016) |
| | | | 1938 | SUMMIT, NEEM, D4 | $NH_4^+$ | Legrand et al. (2016) |
| 1940 | > average $+ 1\sigma$ | No data | | | | |
| 1944 | > average $+ 3\sigma$ | No data | 1942 | NEEM, D4 | $NH_4^+$ | Legrand et al. (2016) |
| | | No data | 1950 | SUMMIT, NEEM, D4 | $NH_4^+$ | Legrand et al. (2016) |
| | | | | SUMMIT | $NH_4^+$, $HCOO^-$ | Savarino and Legrand (1998) |
| | | No data | 1961 | SUMMIT, NEEM, D4 | $NH_4^+$ | Legrand et al. (2016) |
| | | | | SUMMIT | $NH_4^+$, $HCOO^-$ | Savarino and Legrand (1998) |
| | | | | SE-Dome | Levoglucosan | Parvin et al. (2019) |
| | | No data | 1964 | SE-Dome | Levoglucosan | Parvin et al. (2019) |
| | | No data | 1972 | NEEM | rBC | Zennaro et al. (2014) |
| | | | 1973 | NEEM | $NH_4^+$ | Legrand et al. (2016) |
| | | | 1980 | SUMMIT, NEEM, D4 | $NH_4^+$ | Legrand et al. (2016) |
| 1994 | > average $+ 2\sigma$ | No data | 1994 | SE-Dome | Levoglucosan | Parvin et al. (2019) |
| | | | | SUMMIT | Levoglucosan, $NH_4^+$, $HCOO^-$, $CH_3COO^-$, $(C_2H_2O_4)^{2-}$ | Kehrwald et al. (2012), Dibb et al. (1996) |
| 1995 | > average $+ 2\sigma$ | No data | | | | |
| | | No data | 1998 | SE-Dome | Levoglucosan | Parvin et al. (2019) |
| | No CFA data | No data | 2012 | SUMMIT | rBC, $NH_4$ | Keegan et al. (2014) |
| | No CFA data | No data | 2013 | SE-Dome | Levoglucosan | Parvin et al. (2019) |

## Appendix B: Wet deposition vs. dry deposition

We anticipate that the contribution of wet deposition was greater than that of dry deposition. It is difficult to estimate the wet and dry deposition ratio directly, since there are no observations at the SIGMA-D site as at most of the sites in the Arctic. Instead, we estimated the terminal velocity ($V$) of rBC particles falling onto the SIGMA-D site using the equation $V = 2\rho r^2 g / 9\zeta$ assuming spherical rBC particles. Here $\rho$, $r$, $g$, and $\zeta$ denote the density of rBC particles, the radius of rBC particles, the acceleration of gravity, and the viscosity coefficient of the atmosphere, respectively. We used the values $1800\,\mathrm{kg\,m^{-3}}$ and $1\,\mathrm{\mu m}$ for $\rho$ and $2r$ (diameter), respectively. Assuming an atmospheric temperature of $-40\,°\mathrm{C}$, $\zeta$ was calculated to be $1.5 \times 10^{-5}\,\mathrm{N\,s\,m^{-2}}$. With these values, the terminal velocity was estimated to be [TS1] $6\,\mathrm{m\,d^{-1}}$. Given that the rBC particles fall from 500 m above the ice sheet surface at the SIGMA-D site, it would take approximately 100 d for the rBC particles to reach the ice sheet surface, indicating a very small dry deposition velocity at the SIGMA-D site. A study using the GEOS-Chem global chemical transport model (Breider et al., 2014) also indicated that the annual mean fraction of dry deposition in the Arctic was only 11 %.

Furthermore, Sinha et al. (2018) showed that the dry deposition was a small contributor (less than the uncertainties of the measurements, which were about 20 %) to the total rBC deposition at Ny-Ålesund, Svalbard, where the total water-equivalent snowfall amount during September–April was similar to the annual accumulation rate at the SIGMA-D site. Thus, it is reasonable to assume that the contribution of dry deposition is small.

**Data availability.** All the data used in this study are available in the Arctic Data Archive System.

Ice core data, results of air mass back-trajectory analyses, and results of albedo reduction calculations are available at https://doi.org/10.17592/001.2024102301 (Goto-Azuma et al., 2024b).

AWS data are available at https://ads.nipr.ac.jp/dataset/A20241021-001 (Aoki et al., 2024).

**Author contributions.** KGA designed the study and led the article writing. YOT was responsible for the BC measurements. YOT, MH, RD, and JO performed the CFA analyses of the SIGMA-D core. MH and SM measured the ion concentrations in the discrete samples. KoF, SM, AT, and NN dated the SIGMA-D ice core. MoK calculated annual accumulation rates. KGA, YOT, and KaF ana-

lyzed the CFA data. KGA, YOT, NM, TM, SO, YK, and MaK interpreted the BC data. KoF performed backward-trajectory analyses. TA analyzed the AWS data. SM examined the melt features in the SIGMA-D ice core. TA designed and led the ice-coring project at SIGMA-D. TA computed the impacts of BC on albedo. All the authors discussed the results.

**Competing interests.** The contact author has declared that none of the authors has any competing interests.

**Disclaimer.** Publisher's note: Copernicus Publications remains neutral with regard to jurisdictional claims made in the text, published maps, institutional affiliations, or any other geographical representation in this paper. While Copernicus Publications makes every effort to include appropriate place names, the final responsibility lies with the authors.

**Acknowledgements.** We would like to thank Hideaki Motoyama, Tetsuhide Yamasaki, Masahiro Minowa, Yukihiko Onuma, and Yuki Komuro for drilling the SIGMA-D core, processing it in the field, and installing the AWS at the SIGMA-D site. We thank the three anonymous reviewers for their valuable and insightful comments.

**Financial support.** This research has been supported by the Japan Society for the Promotion of Science (JSPS KAKENHI grant nos. JP22221002, JP23221004, and JP18H04140); the Ministry of Education, Culture, Sports, Science and Technology (the Arctic Challenge for Sustainability (ArCS) Project grant no. JPMXD130000000 and the Arctic Challenge for Sustainability II (ArCS II) Project grant no. JPMXD1420318865); and the Environmental Restoration and Conservation Agency (the Environment Research and Technology Development Funds grant nos. JPMEERF20172003, JPMEERF20202003, and JPMEERF20232001).

**Review statement.** This paper was edited by Aurélien Dommergue and reviewed by three anonymous referees.

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

## Remarks from the typesetter