# Peer review of "High-resolution analyses of concentrations and sizes of refractory"

_EGUsphere, 2024_

## Author Comment (AC2)

Response to RC2

We thank Referee 2 for the very helpful and valuable comments. We will take all the comments into consideration and revise our manuscript. Our responses to the Referee's comments are shown below. The Referee's comments and our replies are numbered and shown in blue and black, respectively.

The manuscript entitled

High-resolution analyses of concentrations and sizes of black carbon particles deposited on northwest Greenland over the past 350 years - Part 2: Seasonal and temporal trends in black carbon originated from fossil fuel combustion and biomass burning

by Goto-Azuma et al.,

presents data of concentrations and size distributions of refractory black carbon (rBC) particles from an ice core in the Arctic based on a novel CFA system equipped with a more efficient nebulisation system used to fed a wide-range SP2 system, used for the rBC determination. Overall the system provide an unprecedented temporal resolution and extend the size range of rBC determination.

RC2-1 The manuscript (labelled Part2) discusses trends and sources of BC particles, presents a back trajectory analysis, the identification of potential sources   (biomass burning and fossil fuel combustion) and an estimation of potential albedo reductions.   The results presented in this manuscript rely strongly on the technical details presented in Part 1 paper. I recommend Part2 to be finalised after the acceptance of Part1.

AC2-1 We will revise the Part 1 manuscript first and would like to leave the decision to the Editor.

Major points

RC2-2 The assumptions used to obtain the monthly resolution looks artificial because has been obtained by assuming an equally spaced monthly time step. This imply the assumption of a constant monthly dry and wet deposition rate which is not the case in particular considering the 350 years time span investigated. Furthermore it is not clear how much the signal is blurred by the dispersion in the CFA system. Indeed, the data have a clear and well resolved seasonal signal, which is already

a very nice results in my opinion. The sections of the manuscript where the monthly analysis is presented should be revised.

**AC2-2** We acknowledge that the term "month" in our manuscript does not correspond to a real calendar month, as wet and dry deposition do not occur evenly throughout a year. The use of the word "monthly" may have been confusing. As we explained in our reply (AC1-1) to Referee 1's comment RC1-1, we defined the mid-winter and mid-summer of each year, then divided the periods from mid-winter to mid-summer and mid-summer to mid-winter into six equal sections each. In the manuscript, we referred to each of these sections as a "month." However, there is inherent uncertainty in this definition. The discrepancy between our artificial months and actual calendar months could be one or two months. We will revise the manuscript to clearly state that the "months" defined in this study are not real calendar months and to acknowledge the associated uncertainty.

The signal dispersion is discussed in the companion manuscript (Part 1 of our study). Therefore, we did not explain it in this manuscript. However, to address Referee 2's concern, we will include a brief explanation in our revised manuscript, as we mentioned in our reply (AC1-3) to Referee 1's comment RC1-3.

**RC2-3** Back Trajectories analysis -   The 1958-2015 timeframe was used for BT calculations. This period is not necessarily representative of the entire 350 years covered by the ice core. Also, monthly precipitation data have been used to weight BT's   which is highly uncertain. Finally, dry deposition processes have been neglected in this analysis. Overall this part of the work is weak.

**AC2-3** We agree that the 1958-2015 timeframe used for back trajectory calculations is not necessarily representative of the entire 350-year period. However, we are constrained by the availability of reanalysis data, which begins in 1958. Nagatsuka et al. (CP, 2021), based on the back trajectory studies for the SIGMA-D site, have shown that the contributions from different regions remained relatively constant during 1958-2013. Thus, we can discuss the source regions of BC for this period based on our back trajectory calculations. For earlier periods, we can only hypothesize the back trajectories by assuming similar atmospheric circulations to those of the 1958-2015 period. In our revised manuscript, we will clarify this limitation and acknowledge the significant uncertainties for the period prior to the 1950s.

We did not use monthly precipitation data; instead, we used daily precipitation data, which has much lower uncertainties. For back trajectory calculations, we assumed wet deposition. As stated in our reply (AC1-2) to Referee 1's comment (RC1-2), it is reasonable to assume that the contribution of

dry deposition is minor. However, we have performed new calculations of back trajectories without weighing for daily precipitation. In our revised manuscript, we will present these new results and include a brief discussion on wet and dry deposition.

Minor points:

**RC2-4:** 1-rBC (refractory BC) should be used in place of  BC everywhere in the paper.

**AC2-4** We will replace "BC" with "rBC" in our revised manuscript.

**RC2-5:** 2-line 18 - "roles" is vague. "processes" ?

**AC2-5** We will change the word in our revised manuscript.

**RC2-6:** 3- line 40   - "Changes in". -->     "Increase of"

**AC2-6** We will change the word in our revised manuscript.

**RC2-7:** 4- line 177 - 179 -  Na+ was used for annual layer counting a part cases when this was not possible because of the low signal. Therefore the authors recurred to the standard 18O isotope signal.  Using Na+  is interesting but why 18O was not used if the signal is more clear?  A brief explanation should be given.

**AC2-7** Due to the diffusion of water isotopes, seasonal peaks of $\delta^{18}O$ are often less pronounced compared to those of Na, as illustrated by the $\delta^{18}O$ peak around 16.5 m in Fig.2. Consequently, we primarily relied on Na for annual layer counting. Nonetheless, we also examined the $\delta^{18}O$ data and confirmed that both $\delta^{18}O$ and Na yield the same dating results. We will add a brief explanation on annual layer counting in our revised manuscript.

**RC2-8:** 5- Figure 6 - the meaning of the x axis is not clear. Also, what happen to the IC contribution at the D4 site for low x values?

**AC2-8** We apologize for the missing X-axis title. The X-axis denotes days. We will add the X axis title to Figure 6. At the D4 site, the contribution of IC is large in the initial few days, likely due to its proximity to Greenland. IC is the only land region near Greenland that can serve as a BC source region. Despite its small area, the contributions from IC in the initial few days are much larger than those from other land regions. However, if we include oceanic regions along with land regions, the contribution from IC decreases substantially.

**RC2-9:** 6- Figure 8 the normalisation procedure should be briefly described

**AC2-9** In our revised manuscript, we will provide a brief explanation of how we normalized the size distributions.

**RC2-10:** 7- line 284 - why a Gaussian distribution is assumed? usually a log normal distribution is more appropriate.

**AC2-10** We made an error in our manuscript. In our study, we assumed a log-normal distribution. We will correct this mistake in our revised manuscript.

**RC2-11:** 8-lines 285 -301   how the use of a single MMD or single mBC parameter to describe the entire BC size distribution can be consistent with two very different BC sources ?

**AC2-11** This is an important and intriguing comment. The all-season size distribution patterns of BC particles across different decades were similar (Fig. 8), showing distributions close to the log-normal distribution, with additional large particles beyond the log-normal curves. However, both MMD and mBC increased during periods when the contribution of fossil fuel combustion BC particles was high. To better differentiate between fossil fuel combustion and biomass burning BC, we will present decadal means of summer and winter mean mass size distributions. We tentatively label this new figure as Fig. 8-2. Both summer and winter patterns are close to log-normal distributions, though MMDs were larger in winter during periods with significant contributions from fossil fuel

combustion BC, suggesting that fossil fuel combustion BC particles were larger than those from biomass burning. As Referee 2 commented, using only single MMDs or single mBCs is insufficient to describe the entire BC size distributions. We will include the new figure (Fig. 8-2) and explain this in the revised version of our manuscript.

**RC2-12:** 9- Figure 10 - What I see in these plots are yearly BC mass concentration trends. Not monthly BC concentration trends.

**AC2-12** In Fig. 10, we plotted monthly mean BC mass concentrations (though they might not represent true monthly mean concentrations due to uncertainties in monthly dating), not annual mean concentrations. The figure caption is correct.

**RC2-13:** 10-line 351-353.   the BT analysis was inconclusive and should not mentioned further

**AC2-13** We agree that our back trajectory analysis was inconclusive in explaining anthropogenic BC mass concentration trends. Furthermore, there are uncertainties in back trajectories for the period prior to 1958. Despite these uncertainties, we believe it is still worthwhile to speculate on the sources of biomass burning BC, assuming that the back trajectories for the period prior to 1958 were not significantly different from those in the 1958-2015 period. In the revised version of our manuscript, we will clearly state that we can only speculate on the sources and that we cannot draw firm conclusions.

**RC2-14:** 11- line 366 -   how long the high peaks have really lasted? It is possible to catch strong events shorter than 1 season if "the data for a few months after large BC concentration peaks could have been affected" (as affirmed by the authors at   line 206-207).

**AC2-14** The wording "the data for a few months after large BC concentration peaks could have been affected" might have been misleading. Most of the high peaks lasted for only a month or two (with "month" not being a real calendar month, as "month" is not a real month). We will revise this sentence to avoid any confusion.

---

## Author Comment (AC3)

Response to RC3

We thank Referee 3 for the very helpful and valuable comments. We will take all the comments into consideration and revise our manuscript. Our responses to the Referee's comments are shown below. The Referee's comments and our replies are numbered and shown in blue and black, respectively.

**RC3-1**

This study presents a high-resolution (monthly resolved) black carbon (BC) record covering the past 350 years from an ice core drilled on northwest Greenland. There has been much focus on BC during the past decades because of its impact on the Earth's radiation budget. However, BC in snow and ice can be analyzed with several different methods capturing different particle sizes, thus, there is a problem comparing the records and making conclusions about spatial and temporal variations.

The BC record presented in this paper is based on an improvement of a continuous flow analysis (CFA) system providing both BC particle size and mass/number concentrations. This new method is described in another submitted paper by the first author. The particle size analysis makes it possible to distinguish the impact from anthropogenic and biomass burning. As a results it has now been possible to more accurately pinpoint the timing of anthropogenic impact to this part of Greenland than any of the previous studies of BC in ice core from Greenland which have only based their results on mass concentrations. This is a new and important finding. In addition, the results show a shift in seasonality of the annual concentration peak of BC related to the type of impact. A backward trajectory analysis suggests North America as the main source, in agreement with similar ice core-based studies from Greenland.

The paper is generally well written, and the figures are illustrative. ***However, I consider the lack of incorporation of the accumulation record in the discussion of the temporal variability of the BC a problem. The accumulation variability will have a direct impact on the BC transport and deposition and I recommend that this is included in the paper.***

Overall, this is an important study that clearly demonstrates how ice cores and the continuous work in development of different analytical methods can help in understanding the long-range transport of BC to the Arctic.

**AC3-1** We understand Referee 3's concern. In our revised manuscript, we will include the annual accumulation record and the annual BC mass flux calculated using it (Fig. A3). Since there are no apparent long-term trends in annual accumulation rates, the temporal trends in BC mass

concentrations and BC mass fluxes are consistent. We will add Fig. A3 as an Appendix while continuing to use BC concentration data in the main text.

[Figure]

**Figure A3: Annual mean and decadal mean BC mass concentrations (red) together with annual accumulation rate (blue) and annual BC mass flux (green).**

Specific comments

**RC3-2** Line 182-187: I appreciate the careful explanation of the dating, but nature is complex. To have distinct seasonal variability there must be precipitation all year around and the lack of such information in the papers makes it difficult to fully assess the reliability of this statement. Is there any AWS nearby? Maybe this information is included in one of the previous papers, but I think that this must be included in this manuscript since one of the main conclusions is the change of seasonality in the BC input depending on biomass burning or anthropogenic sources.

**AC3-2** As we explained in our replies (AC1-1 and AC2-2) to Referees 1's and 2's comments ( RC1-2 and RC2-2, respectively), our definition of "months" does not correspond to real calendar months. We will present the snow depth data from AWS in Fig. A4. The AWS experienced some issues, and we can only use data from a limited period (May 2014 – September 2015). It appears that there was more precipitation in summer than in winter at the SIGMA-D site, which could introduce some bias in monthly dating. However, precipitation occurred in all months, and the seasonal variation in

precipitation seems to exhibit significant year-to-year variability. By averaging monthly mean concentrations over 10-20 years (Fig. 11), we believe we can observe changes in the seasonality of BC.

[Figure]

**Figure A4: Monthly snow height change at the SIGMA-D site observed with an automatic weather station.**

**RC3-3** Line 196: The study is assuming only wet deposition of BC. What is that based on?

**AC3-3** When we calculated back trajectories, we assumed only wet deposition of BC for the reasons outlined in our replies (AC1-2 and AC2-3) to Referee 1's comment (RC1-2) and Referee 2's comment (RC2-3). In our revised manuscript, we will explain this reasoning and include the results of the back trajectory calculations without assuming wet deposition, as shown in AC1-2. We will also clarify that our interpretation of the ice core data would not change if there is a minor contribution from dry deposition.

**RC3-4** Line 303. In this paragraph it becomes evident that the temporal variability of accumulation is not included in the discussion in this paper. As already mentioned, I think this is such a fundamental part of the interpretation of the BC record that it cannot be left out.

**AC3-4** We agree. Please refer to AC3-1 for further details.

**RC3-5** With the mean annual temperature of -23ºC I assume that there are not any melt layers that is disturbing the stratigraphy but there could still be ice lenses created by solar radiation during the summer. Is that the case here?

**AC3-5** We examined the melt features (ice layers and thin crusts) in the uppermost 20 meters of the SIGMA-D ice core, where increased summer melting would be expected due to recent warming. We observed occasional ice layers, with a maximum thickness of 10 mm at only three depths. The 20-meter average melt feature percentage (MFP) was just 0.47%. The maximum MFP per meter was 1.7%, and 10 out of the 20 meters had no melt features at all. Thus, the effects of melt-refreeze cycles are minimal at the SIGMA-D site. We will briefly explain this in our revised manuscript.

---

## Author Comment (AC4)

Response to RC1

We thank Referee 1 for the very helpful and valuable comments. We will take all the comments into consideration and revise our manuscript. Our responses to the Referee's comments are shown below. The Referee's comments and our replies are numbered and shown in blue and black, respectively.

**RC1-1** *This manuscript "High-resolution analyses of concentrations and sizes of black carbon particles deposited on northwest Greenland over the past 350 years – Part 2: Seasonal and temporal trends in black carbon originated from fossil fuel combustion and biomass burning" submitted by Goto-Azuma et al., provided monthly resolved 350-year records of concentrations and size distributions of black carbon (BC) particles from an ice core that was drilled in the northwest Greenland Ice Sheet. The authors discussed sources of BC particles originated from biomass burning and fossil fuel combustion based on backward trajectory analyses, and estimated the potential albedo reductions. The main advantage of this work concerns with the extremely high resolution records from the updated CFA system that was consisted of single-particle soot photometer and a high-efficiency nebulizer. **As a result, the annual layer of the SIGMA-D ice core can be reasonably divided into 12 months, which provides a chance to decipher the monthly variations of the ice core BC particles that have been impossible before. However, these monthly-averages depend on assumptions, such as evenly month distribution of precipitation, that can be hardly met. Therefore, uncertainties due to the assumptions should be examined carefully before a solid conclusion can be reached.***

**AC1-1** We have identified the mid-winter and mid-summer points of each year and divided the period from mid-winter to mid-summer into six equal sections. Similarly, the period from mid-summer to mid-winter was also divided into six equal sections. In the manuscript, we referred to each of these sections as a "month." However, we acknowledge that these sections do not correspond to real calendar months, as precipitation does not occur evenly throughout the year. The "months" defined in this manuscript are artificial. Nonetheless, we think that dividing each year into 12 sections provides valuable insights into the seasonality of BC and its temporal changes. We will revise the manuscript to clearly state that the "months" defined in this study are not real calendar months and to acknowledge the associated uncertainties.

*Other minor comments:*

**RC1-2** *Lines 196-197: Please give pieces of evidence that contribution of dry deposition can be ignored.*

**AC1-2** When we computed back trajectories, we assumed that the contribution of wet deposition was greater than that of dry deposition. It is difficult to estimate the wet and dry deposition ratio directly since there are no observations at the SIGMA-D site, as at most of the sites in the Arctic. Instead, we computed the terminal velocity (V) of BC particles falling onto the SIGMA-D site using the equation $V=2\rho r^2 g/9\zeta$ assuming spherical BC particles. Here $\rho$, r, g, and $\zeta$ denote the density of BC particles, the radius of BC particles, the acceleration of gravity, and the viscosity coefficient of the atmosphere, respectively. We used the values 1800 kg m$^{-3}$ and 1 μm for $\rho$ and 2r (diameter), respectively. Assuming an atmospheric temperature of -40$^{\circ}$C, $\zeta$ was calculated to be $1.5\times10^{-5}$ N s m$^{-2}$. With these values, the terminal velocity was estimated to be 6 m day$^{-1}$. Given that the BC particles fall from 500 m above the ice sheet surface at the SIGMA-D site, it would take approximately 100 days for the BC particles to reach the ice sheet surface, indicating a very small dry deposition velocity at the SIGMA-D site. A study using the GEOS-Chem global chemical transport model (Breider et al., JGR, 2014) also indicated that the annual mean fraction of dry deposition in the Arctic was only 11 %. Furthermore, Sinha et al. (JGR, 2018) showed that the dry deposition was a small contributor (less than the uncertainties of the measurements, which were about 20%) to the total BC deposition at Ny-Ålesund, Svalbard, where the total water equivalent snowfall amount during September-April was similar to the annual accumulation amount at the SIGMA-D site. Thus, it is reasonable to assume that the contribution of dry deposition is small. We will add these statements when we revise our manuscript.

We also conducted additional back trajectory analyses without weighting the air mass probability by local daily precipitation. The probability distribution of air masses (Fig. A2) indicates that, throughout the year, back trajectories for the SIGMA-D and D4 sites originate from wider regions without weighting, with the difference being more pronounced in winter than in summer. Fig. 6_new compares the results with and without weighting. Whether weighted by daily precipitation or not, the contribution from North America is the largest at both SIGMA-D and D4 sites throughout the year. Without weighting, the contribution from Russia increases at both sites throughout the year, with a more significant increase at the SIGMA-D site, especially in winter. Without weighting, the contribution from Europe decreases in winter and increases in summer at both sites (at SIGMA-D after six days in winter). The slight difference in temporal patterns of mass concentrations of anthropogenic BC between the two sites might reflect the different contributions from Russia in winter. We will replace Fig. 6 with Fig. 6_new and add Fig. A2 in the revised manuscript. Our interpretation of the ice core data remains unchanged if there is a minor contribution from dry deposition.

[Figure]

**Fig. A2: Probability distributions of air masses at (left) the SIGMA-D site and (right) the D4 site calculated without weighing with the local daily precipitation: (a) averages of all seasons, (b) averages of winter months (December-February), and (c) averages of summer months (May-July)**

[Figure]

**Fig. 6_new: Temporal variability in contribution of air masses arriving at (left) the SIGMA-D site and (right) the D4 site from four regions: (a) averages of 12 months, (b) averages of winter months (December-February), and (c) averages of summer months (May-July). Right-hand axes indicate contributions from NA, and left-hand axes indicate contributions from the other regions. Solid and dotted curves denote results with and without weighing with the daily local precipitation, respectively.**

**RC1-3** *Section 3.2. Is there possibility that corresponding of the seasonal Na and BC peaks (Fig. 10) can be disturbed by signal dispersion in the CFA system?*

**AC1-3** Figure 10 shows Na and BC concentrations at ~ 1/12 year resolution, which corresponds to ~ 22 mm resolution. As demonstrated in the companion paper (Part 1 of our study), the dispersion lengths are ~35 and ~39 mm for Na and BC, respectively. Consequently, the signal dispersion might slightly reduce the heights of the seasonal peaks. Due to the slightly asymmetrical shape of the Na and BC peaks (as discussed in the companion paper), their positions might change slightly in the CFA system. However, these change in peak positions would be minimal and would not alter the relative positions of the seasonal Na and BC peaks.

**RC1-4** *Figure 3: The extremely high peak around 1710 needs to be explained.*

**AC1-4** The extremely high BC peak in 1710 occurred in summer and was accompanied by a very high $NH_4^+$ peak. Similarly, other very high BC peaks in 1711 and 1712 occurred in summer and were accompanied by high $NH_4^+$ peaks. These observations suggest that these BC peaks likely originated from large forest fires. As suggested by Referee 1 commented, we will add a few sentences highlighting these peaks in the revised version of our manuscript.

**RC1-5** *Line 257: "GriIS" → "GrIS"*

**AC1-5** We will correct this typo.

**RC1-6** *Figure 6: Please indicate the meaning of the X axis.*

**AC1-6** We apologize for the missing X-axis title. The X-axis represents days. We will add the X-axis title in the revised version.

**RC1-7** *Line 291: "Of the two size parameters, mBC is easier to calculate than MMD; hence, it can be used to investigate changes with high temporal resolution". This is not a good reason to choose mBC.*

**AC1-7** We fully agree. This does not sound scientific. We will revise this sentence.

**RC1-8** *Lines 300-301: An objective statistical method should be applied to reach the results.*

**AC1-8** To address Referee comment RC1-8, we performed breakpoint analyses (Muggeo, Statis. Med., 2003). The results suggest that the increases of BC mass concentrations, mBC and MMD began around the 1850s, 1870s and 1810s, respectively. Although these dates are slightly different from those stated in our manuscript, the increases in mBC and MMD still occurred earlier than the increase in concentrations, as we originally indicated. In the revised manuscript, we will incorporate the results of the breakpoint analyses.

**RC1-9** *Figure 10: The time range of Figure 10C is not 1915-2003.*

**AC1-9** We apologize for this oversight. We will correct it promptly.

---

## Author Response (AR1)

We thank Referees 1, 2, and 3 for their thorough reviews and insightful comments, which have significantly improved the manuscript. Our responses to the Referee's comments are shown below. The Referee's comments and our replies are numbered and shown in blue and black, respectively.

**Response to RC1**

We thank Referee 1 for the very helpful and valuable comments. We have taken all the comments into consideration and revised our manuscript. Our responses to the Referee's comments are shown below.

**RC1-1** *This manuscript "High-resolution analyses of concentrations and sizes of black carbon particles deposited on northwest Greenland over the past 350 years – Part 2: Seasonal and temporal trends in black carbon originated from fossil fuel combustion and biomass burning" submitted by Goto-Azuma et al., provided monthly resolved 350-year records of concentrations and size distributions of black carbon (BC) particles from an ice core that was drilled in the northwest Greenland Ice Sheet. The authors discussed sources of BC particles originated from biomass burning and fossil fuel combustion based on backward trajectory analyses, and estimated the potential albedo reductions. The main advantage of this work concerns with the extremely high resolution records from the updated CFA system that was consisted of single-particle soot photometer and a high-efficiency nebuliser. As a result, the annual layer of the SIGMA-D ice core can be reasonably divided into 12 months, which provides a chance to decipher the monthly variations of the ice core BC particles that have been impossible before. However, these monthly-averages depend on assumptions, such as evenly month distribution of precipitation, that can be hardly met. Therefore, uncertainties due to the assumptions should be examined carefully before a solid conclusion can be reached.*

**AC1-1** We have identified the mid-winter and mid-summer points of each year and divided the period from mid-winter to mid-summer into six equal sections. Similarly, the period from mid-summer to mid-winter was also divided into six equal sections. In the manuscript, we referred to each of these sections as a "month." However, we acknowledge that these sections do not correspond to real calendar months, as precipitation does not occur evenly throughout the year. The "months" defined in this manuscript are artificial. Nonetheless, we think that dividing each year into 12 sections provides valuable insights into the seasonality of BC and its temporal changes. We have revised the manuscript to clearly state that the "months" defined in this study were not real calendar months and to acknowledge the associated uncertainties.

*Other minor comments:*

**RC1-2** *Lines 196-197: Please give pieces of evidence that contribution of dry deposition can be ignored.*

**AC1-2** When we computed back trajectories, we assumed that the contribution of wet deposition was greater than that of dry deposition. It is difficult to estimate the wet and dry deposition ratio directly since there are no observations at the SIGMA-D site, as at most of the sites in the Arctic. Instead, we estimated the terminal velocity (V) of rBC particles falling onto the SIGMA-D site using the equation $V=2\rho r^2 g/9\zeta$ assuming spherical rBC particles. Here $\rho$, $r$, $g$, and $\zeta$ denote the density of rBC particles, the radius of rBC particles, the acceleration of gravity, and the viscosity coefficient of the atmosphere, respectively. We used the values 1800 kg m$^{-3}$ and 1 μm for $\rho$ and $2r$ (diameter), respectively. Assuming an atmospheric temperature of -40$^{\circ}$C, $\zeta$ was calculated to be $1.5 \times 10^{-5}$ N s m$^{-2}$. With these values, the terminal velocity was estimated to be 6 m day$^{-1}$. Given that the rBC particles fall from 500 m above the ice sheet surface at the SIGMA-D site, it would take approximately 100 days for the rBC particles to reach the ice sheet surface, indicating a very small dry deposition velocity at the SIGMA-D site. A study using the GEOS-Chem global chemical transport model (Breider et al., JGR, 2014) also indicated that the annual mean fraction of dry deposition in the Arctic was only 11 %. Furthermore, Sinha et al. (JGR, 2018) showed that the dry deposition was a small contributor (less than the uncertainties of the measurements, which were about 20%) to the total rBC deposition at Ny-Ålesund, Svalbard, where the total water equivalent snowfall amount during September-April was similar to the annual accumulation amount at the SIGMA-D site. Thus, it is reasonable to assume that the contribution of dry deposition is small. We have added these statements as Appendix B.

We also performed additional back trajectory analyses without weighting the air mass probability by local daily precipitation. We replaced Fig. 6 with a new one, which compares the results with and without weighting. We also added Fig. A5, which shows the probability distributions calculated without weighing with precipitation. The results show that our interpretation of the ice core data remains unchanged, even if there is a minor contribution from dry deposition. In the revised manuscript, we have added a few sentences to compare the precipitation-weighted and unweighted trajectories.

**RC1-3** *Section 3.2. Is there possibility that corresponding of the seasonal Na and BC peaks (Fig. 10) can be disturbed by signal dispersion in the CFA system?*

**AC1-3** Figure 10 shows Na and BC concentrations at ~ 1/12 year resolution, corresponding to ~ 22 mm resolution. As demonstrated in the companion paper (Part 1 of our study), the dispersion lengths are ~35 and ~39 mm for Na and BC, respectively. Consequently, the signal dispersion might slightly reduce the heights of the seasonal peaks. Due to the slightly asymmetrical shape of the Na and BC peaks (as discussed in the companion paper), their positions might change slightly in the CFA system. However, the changes in peak positions would be minimal and would not alter the relative positions of the seasonal Na and BC peaks.

**RC1-4** *Figure 3: The extremely high peak around 1710 needs to be explained.*

**AC1-4** The extremely high BC peak in 1710 occurred in summer and was accompanied by a very high $NH_4^+$ peak. Similarly, other very high BC peaks in 1711 and 1712 occurred in summer and were accompanied by high $NH_4^+$ peaks. These observations suggest that these BC peaks likely originated from large forest fires. As suggested by Referee 1, we added a few sentences explaining these peaks in the revised manuscript.

**RC1-5** *Line 257: "GriIS" → "GrIS"*

**AC1-5** We corrected this typo.

**RC1-6** *Figure 6: Please indicate the meaning of the X axis.*

**AC1-6** We apologise for the missing X-axis title. The X-axis represents days. We have added the X-axis title.

**RC1-7** *Line 291: "Of the two size parameters, mBC is easier to calculate than MMD; hence, it can be used to investigate changes with high temporal resolution". This is not a good reason to choose mBC.*

**AC1-7** We fully agree. This does not sound scientific. We have deleted this sentence.

**RC1-8** *Lines 300-301: An objective statistical method should be applied to reach the results.*

**AC1-8** To address Referee comment RC1-8, we performed breakpoint analyses (Muggeo, Statis. Med., 2003). The results suggest that the increases of BC mass concentrations, mBC and MMD began around the 1870s, 1850s and 1820s, respectively. Although these dates slightly differ from those stated in our initial manuscript, the increases in mBC and MMD still occurred earlier than the increase in concentrations, as we indicated initially. In the revised manuscript, we incorporated the results of the breakpoint analyses.

**RC1-9** *Figure 10: The time range of Figure 10C is not 1915-2003.*

**AC1-9** We apologise for this error. We have corrected it.

**Response to RC2**

We thank Referee 2 for the very helpful and valuable comments. We have taken all the comments into consideration and revised our manuscript. Our responses to the Referee's comments are shown below.

The manuscript entitled

High-resolution analyses of concentrations and sizes of black carbon particles deposited on northwest Greenland over the past 350 years - Part 2: Seasonal and temporal trends in black carbon originated from fossil fuel combustion and biomass burning

by Goto-Azuma et al.,

presents data of concentrations and size distributions of refractory black carbon (rBC) particles from an ice core in the Arctic based on a novel CFA system equipped with a more efficient nebulisation system used to fed a wide-range SP2 system, used for the rBC determination. Overall the system provide an unprecedented temporal resolution and extend the size range of rBC determination.

**RC2-1** The manuscript (labelled Part2) discusses trends and sources of BC particles, presents a back trajectory analysis, the identification of potential sources   (biomass burning and fossil fuel combustion) and an estimation of potential albedo reductions.   The results presented in this manuscript rely strongly on the technical details presented in Part 1 paper. I recommend Part2 to be finalised after the acceptance of Part1.

**AC2-1** Following this suggestion, we have revised the Part 1 manuscript first. The Part 1 paper has been accepted.

Major points

**RC2-2** The assumptions used to obtain the monthly resolution looks artificial because has been obtained by assuming an equally spaced monthly time step. This imply the assumption of a constant monthly dry and wet deposition rate which is not the case in particular considering the 350 years time span investigated. Furthermore it is not clear how much the signal is blurred by the dispersion in the CFA system. Indeed, the data have a clear and well resolved seasonal signal, which is already a very nice results in my opinion. The sections of the manuscript where the monthly analysis is presented should be revised.

 **AC2-2** We acknowledge that the term "month" in our manuscript does not correspond to an actual calendar month, as wet and dry deposition do not occur evenly throughout a year. The use of the word "monthly" may have been confusing. As we explained in our reply (AC1-1) to Referee 1's

comment RC1-1, we defined the mid-winter and mid-summer of each year, then divided the periods from mid-winter to mid-summer and mid-summer to mid-winter into six equal sections each. We referred to each section as a "month." However, there is inherent uncertainty in this definition. The discrepancy between our artificial month and actual calendar month could be one or two months. In the revised manuscript, we stated that the "months" defined in this study were not real calendar months and acknowledged the associated uncertainty.

The signal dispersion is discussed in the companion manuscript (Part 1 of our study). Therefore, we did not explain it in this manuscript. However, to address Referee 2's concern, we included a brief explanation in our revised manuscript, as we mentioned in our reply (AC1-3) to Referee 1's comment RC1-3.

**RC2-3** Back Trajectories analysis -   The 1958-2015 timeframe was used for BT calculations. This period is not necessarily representative of the entire 350 years covered by the ice core. Also, monthly precipitation data have been used to weight BT's   which is highly uncertain. Finally, dry deposition processes have been neglected in this analysis. Overall this part of the work is weak.

**AC2-3** We agree that the 1958-2015 timeframe used for back trajectory calculations is not necessarily representative of the entire 350-year period. However, we are constrained by the availability of reanalysis data, which began in 1958. Nagatsuka et al. (CP, 2021), based on the back trajectory studies for the SIGMA-D site, have shown that the interannual variability of contributions from different regions remained relatively constant during 1958-2013. Thus, we can discuss the source regions of BC for this period based on our back trajectory calculations. For earlier periods, we can only hypothesise the back trajectories by assuming similar atmospheric circulations to those of the 1958-2015 period. In our revised manuscript, we explained this limitation and acknowledged the large uncertainties for the period prior to the 1950s.

We did not use monthly precipitation data; instead, we used daily precipitation data, which has much lower uncertainties. For back trajectory calculations, we assumed wet deposition. As stated in our reply (AC1-2) to Referee 1's comment (RC1-2), it is reasonable to assume that the contribution of dry deposition is minor. However, we have performed new calculations of back trajectories without weighing for daily precipitation. In our revised manuscript, we presented these new results and included a brief discussion on wet and dry deposition.

**RC2-4:** 1-rBC (refractory BC) should be used in place of BC everywhere in the paper.

**AC2-4** We replaced "BC" with "rBC" in the revised manuscript.

**RC2-5:** 2-line 18 - "roles" is vague. "processes" ?

**AC2-5** We have changed "play important roles" to "affect", since "processes" is not what we meant.

**RC2-6:** 3- line 40 - "Changes in". --> "Increase of"

**AC2-6** We have corrected the words.

**RC2-7:** 4- line 177 - 179 - Na+ was used for annual layer counting a part cases when this was not possible because of the low signal. Therefore the authors recurred to the standard 18O isotope signal. Using Na+ is interesting but why 18O was not used if the signal is more clear? A brief explanation should be given.

**AC2-7** Due to the diffusion of water isotopes, seasonal peaks of $\delta^{18}O$ are often less pronounced compared to those of Na, as illustrated by the $\delta^{18}O$ peak around 16.5 m in Fig.2. Consequently, we primarily relied on Na for annual layer counting. Nonetheless, we also examined the $\delta^{18}O$ data and confirmed that both $\delta^{18}O$ and Na yield the same dating results. We added a brief explanation of annual layer counting in our revised manuscript.

**RC2-8:** 5- Figure 6 - the meaning of the x axis is not clear. Also, what happen to the IC contribution at the D4 site for low x values?

**AC2-8** We apologise for the missing X-axis title. The X-axis denotes days. We added the X axis title to Figure 6. At the D4 site, the contribution of IC is significant in the initial few days, likely due to its proximity to Greenland. IC is the only land region near Greenland that can serve as a BC source region. Despite its small area, the contributions from IC in the initial few days are much more significant than those from other land regions. However, if we include oceanic regions along with land regions, the contribution from IC decreases substantially. We added these explanations.

**RC2-9:** 6- Figure 8 the normalisation procedure should be briefly described

**AC2-9** We provided a brief explanation of normalisation in the caption of Fig.8.

**RC2-10:** 7- line 284 - why a Gaussian distribution is assumed? usually a log normal distribution is more appropriate.

**AC2-10** We made an error in our manuscript. In our study, we assumed a log-normal distribution. We corrected this mistake in the revised manuscript.

**RC2-11:** 8-lines 285 -301   how the use of a single MMD or single mBC parameter to describe the entire BC size distribution can be consistent with two very different BC sources ?

**AC2-11** This is an important and intriguing comment. The all-season size distribution patterns of rBC particles across different decades were similar (Fig. 8), showing distributions close to the log-normal distribution, with additional large particles beyond the log-normal curves. However, both MMD and mBC increased during periods when the contribution of fossil fuel combustion rBC particles was high. As Referee 2 commented, using only single MMDs or single mBCs is insufficient to describe the entire rBC size distributions. To better differentiate between fossil fuel combustion and biomass burning rBC, we added a new figure (Fig. 13 in the revised manuscript) showing decadal means of summer and winter mean mass size distributions. Both summer and winter patterns are close to log-normal distributions, though MMDs were larger in winter during periods with significant contributions from fossil fuel combustion rBC, suggesting that fossil fuel combustion rBC particles were larger than those from biomass burning.

**RC2-12:** 9- Figure 10 - What I see in these plots are yearly BC mass concentration trends. Not monthly BC concentration trends.

**AC2-12** In Fig. 10, we plotted monthly mean BC mass concentrations (though they might not represent true monthly mean concentrations due to uncertainties in monthly dating), not annual mean concentrations. The figure caption is correct.

**RC2-13:** 10-line 351-353.   the BT analysis was inconclusive and should not mentioned further

**AC2-13** We agree that our back trajectory analysis was inconclusive in explaining anthropogenic BC mass concentration trends. Furthermore, there are uncertainties in back trajectories for the period prior to 1958. Despite these uncertainties, we believe it is still worthwhile to speculate on the sources of biomass burning BC, assuming that the back trajectories for the period prior to 1958 were not significantly different from those in the 1958-2015 period. In the revised manuscript, we stated that there were large uncertainties.

**RC2-14:** 11- line 366 -   how long the high peaks have really lasted? It is possible to catch strong events shorter than 1 season if "the data for a few months after large BC concentration peaks could have been affected" (as affirmed by the authors at   line 206-207).

**AC2-14** The wording "the data for a few months after large BC concentration peaks could have been affected" might have been misleading. Most of the high peaks lasted for only a month or two (with "month" not being a real calendar month, as "month" is not an actual month). We revised Line 206-207 and Line 366, where Line numbers are those in the initial manuscript.

**Response to RC3**

We thank Referee 3 for the very helpful and valuable comments. We have taken all the comments

into consideration and revised our manuscript. Our responses to the Referee's comments are shown below.

**RC3-1**

This study presents a high-resolution (monthly resolved) black carbon (BC) record covering the past 350 years from an ice core drilled on northwest Greenland. There has been much focus on BC during the past decades because of its impact on the Earth's radiation budget. However, BC in snow and ice can be analysed with several different methods capturing different particle sizes, thus, there is a problem comparing the records and making conclusions about spatial and temporal variations.

The BC record presented in this paper is based on an improvement of a continuous flow analysis (CFA) system providing both BC particle size and mass/number concentrations. This new method is described in another submitted paper by the first author. The particle size analysis makes it possible to distinguish the impact from anthropogenic and biomass burning. As a results it has now been possible to more accurately pinpoint the timing of anthropogenic impact to this part of Greenland than any of the previous studies of BC in ice core from Greenland which have only based their results on mass concentrations. This is a new and important finding. In addition, the results show a shift in seasonality of the annual concentration peak of BC related to the type of impact. A backward trajectory analysis suggests North America as the main source, in agreement with similar ice core-based studies from Greenland.

The paper is generally well written, and the figures are illustrative. *__However, I consider the lack of incorporation of the accumulation record in the discussion of the temporal variability of the BC a problem. The accumulation variability will have a direct impact on the BC transport and deposition and I recommend that this is included in the paper.__*

Overall, this is an important study that clearly demonstrates how ice cores and the continuous work in development of different analytical methods can help in understanding the long-range transport of BC to the Arctic.

**AC3-1** We understand Referee 3's concern. In the revised manuscript, we added the annual accumulation record and the annual BC mass flux calculated using it (Fig. A4). Since there are no apparent long-term trends in annual accumulation rates, the temporal trends in rBC mass

concentrations and rBC mass fluxes are consistent. Therefore, we used concentration data for further discussion. We added these explanations in the text.

Specific comments

**RC3-2** Line 182-187: I appreciate the careful explanation of the dating, but nature is complex. To have distinct seasonal variability there must be precipitation all year around and the lack of such information in the papers makes it difficult to fully assess the reliability of this statement. Is there any AWS nearby? Maybe this information is included in one of the previous papers, but I think that this must be included in this manuscript since one of the main conclusions is the change of seasonality in the BC input depending on biomass burning or anthropogenic sources.

**AC3-2** As we explained in our replies (AC1-1 and AC2-2) to Referees 1's and 2's comments ( RC1-2 and RC2-2, respectively), our definition of "months" does not correspond to real calendar months. We presented the snow depth data from an AWS installed at the SIGMA-D site in Figs. A1 and A2. The AWS experienced some issues, and we can only use data from a limited period (May 2014 – September 2015). There appears to be more precipitation in summer than in winter at the SIGMA-D site, which could introduce some bias in monthly dating. Although the AWS data indicated that precipitation occurred in all months (Fig. A2), there were a few months when snow height change was negative, likely mainly owing to wind scouring. Moreover, the seasonal variation in precipitation seems to exhibit significant year-to-year variability. However, by averaging monthly mean concentrations over 10-20 years (Fig. 11), we believe we can observe changes in the seasonality of rBC.

**RC3-3** Line 196: The study is assuming only wet deposition of BC. What is that based on?

**AC3-3** When we calculated back trajectories, we assumed only wet deposition of BC for the reasons outlined in our replies (AC1-2 and AC2-3) to Referee 1's comment (RC1-2) and Referee 2's comment (RC2-3). In the revised manuscript, we explained this reasoning and included the results of the back trajectory calculations without assuming wet deposition, as mentioned in AC1-2. We also stated that our interpretation of the ice core data would not change even if there was a minor contribution from dry deposition.

**RC3-4** Line 303. In this paragraph it becomes evident that the temporal variability of accumulation is not included in the discussion in this paper. As already mentioned, I think this is such a fundamental part of the interpretation of the BC record that it cannot be left out.

**AC3-4** We agree. Please refer to AC3-1 for further details.

**RC3-5** With the mean annual temperature of -23°C I assume that there are not any melt layers that is disturbing the stratigraphy but there could still be ice lenses created by solar radiation during the summer. Is that the case here?

**AC3-5** We examined the melt features (ice layers and thin crusts) in the uppermost 20 meters of the SIGMA-D ice core, where increased summer melting would be expected due to recent warming. We observed occasional ice layers, with a maximum thickness of 10 mm at only three depths. The 20-meter average melt feature percentage (MFP) was just 0.47%. The maximum MFP per meter was 1.7%, and 10 out of the 20 meters had no melt features at all. Thus, the effects of melt-refreeze cycles are minimal at the SIGMA-D site. We have explained this in the revised manuscript.

**Additional authors' changes in the manuscript**

In the initial manuscript, we assumed that the maximum measurable rBC diameter was 650 nm for the off-the-shelf SP2. However, a referee who reviewed the companion paper (i.e. Part 1 of our study) pointed out that it was 500 nm, not 650 nm. Accordingly, we changed Fig.4.

Additionally, we have also made minor editorial changes, such as corrections of errors (including those in Figures) and minor changes in English. Also, we have added two coauthors, and slightly modified author contributions and acknowledgements.